

# Hydrological assessment of atmospheric forcing uncertainty in the Euro-Mediterranean area using a land surface model

Emiliano Gelati[1,*], Bertrand Decharme[1], Jean-Christophe Calvet[1], Marie Minvielle[1], Jan Polcher[2], David Fairbairn[1], Graham P. Weedon[3]

[1]CNRM, UMR 3589 (Météo-France, CNRS), Toulouse, France

[2]Laboratoire de Météorologie Dynamique du CNRS, UMR8539 (IPSL, CNRS), Paris, France

[3]Met Office, Joint Centre for Hydrometeorological Research, Wallingford, United Kingdom

[*]Now at: Joint Research Centre, European Commission, Ispra, Italy

*Correspondence to*: Jean-Christophe Calvet (jean-christophe.calvet@meteo.fr)

**Abstract.** The understanding of land surface hydrology is critical for planning human activities involving freshwater resources. We assess how atmospheric forcing data uncertainties affect land surface model (LSM) simulations by means of an extensive evaluation exercise using a number of state-of-the-art remote sensing and station-based datasets. For this purpose, we use the CO2-responsive ISBA-A-gs LSM coupled the CNRM version of the Total Runoff Integrated Pathways (CTRIP) river routing model. We perform multi-forcing simulations over the Euro-Mediterranean area (25°–75.5°N, 11.5°W–62.5°E, at 0.5° resolution) from 1979 to 2012. The model is forced using four atmospheric datasets. Three of them are based on the ERA-Interim reanalysis (ERA-I). The fourth dataset is independent from ERA-Interim: PGF, developed at Princeton University. The hydrological impacts of atmospheric forcing uncertainties are assessed by comparing simulated surface soil moisture (SSM), leaf area index (LAI) and river discharge against observation-based datasets: SSM from the European Space Agency's Water Cycle Multi-mission Observation Strategy and Climate Change Initiative projects (ESA-CCI); LAI of the Global Inventory Modeling and Mapping Studies (GIMMS); and Global Runoff Data Centre (GRDC) river discharge. The atmospheric forcing data are also compared to reference datasets. Precipitation is the most uncertain forcing variable across datasets, while the most consistent are air temperature, and SW and LW radiation. At the monthly time scale, SSM and LAI simulations are relatively insensitive to forcing uncertainties. Some discrepancies with ESA-CCI appear to be forcing-independent and may be due to different assumptions underlying the LSM and the remote sensing retrieval algorithm. All simulations overestimate average summer and early autumn LAI. Forcing uncertainty impacts on simulated river discharge are larger on mean values and standard deviations than on correlations with GRDC data. Anomaly correlation coefficients are not inferior to those computed from raw monthly discharge time series, indicating that the model reproduces inter-annual variability fairly well. However, simulated river discharge time series generally feature larger variability compared to measurements. They also tend to overestimate winter-spring high flows and underestimate summer-autumn low flows. Considering that several differences emerge between simulations and reference data, which may not be completely explained by forcing uncertainty, we suggest several research directions. These range from further investigating the



discrepancies between LSMs and remote sensing retrievals to developing new model components to represent physical and anthropogenic processes.

## 1 Introduction

Freshwater resources are at the core of primary human needs. In particular, the production and supply of food and energy are
closely interconnected with water availability and quality (Bazilian et al., 2011; Ringler et al., 2013; Lawford et al., 2013; Damerau et al., 2016). Climate change may add further constraints to the sustainable use of water resources, by causing increased drought frequency and intensity in vulnerable regions such as the Euro-Mediterranean area (Planton et al., 2012; IPCC, 2014). Therefore, understanding and predicting water cycle processes on continental surfaces are necessary to build planning and assessment tools for integrated water resources management.

Large-scale hydrology can be simulated using several approaches, ranging from lumped water balance models to distributed land surface models (LSMs), which can be coupled to atmospheric general circulation models (AGCMs; Widén-Nilsson et al., 2007). LSMs can also be run off-line, forced by gridded atmospheric datasets that are generally obtained from AGCM reanalysis or simulation. LSM and large-scale hydrological simulations are used to study a wide range of water-related problems: hydrological and agricultural droughts (Dirmeyer et al., 2006; Szczypta et al., 2012, 2014); floods (Decharme et
al., 2012; Pappenberger et al., 2012; Hirpa et al., 2016); agricultural production and irrigation (Rost et al., 2009; Jägermeyr et al., 2016); surface freshwater temperature and its impact on energy production (van Beek et al., 2012; van Vliet et al., 2012; Yearsley, 2012).

The lack of direct observations suitable for large-scale quantification of spatially heterogeneous hydrological variables (e.g. soil moisture and evapotranspiration) gives LSMs a key role in water resources assessments. However, to obtain reliable
hydrological simulations from LSMs, several uncertainties need to be tackled. Their main sources are: model physics (Dirmeyer, 2011; Beck et al., 2016); land surface parameters (Douville, 1998; Milly and Shmakin 2002); and atmospheric forcing (Ngo-Duc et al., 2005; Decharme and Douville, 2006; Nasonova et al., 2011). The experiments of Guo et al. (2006b) found that atmospheric forcing uncertainties affect LSM hydrological simulations as much as uncertainties stemming from the models themselves.

Uncertainty impacts can be assessed by comparing simulation outputs with available independent datasets of land surface state variables and fluxes. Such datasets should be representative at the current spatial scales of LSMs (101–102 km).

River discharge is useful for validating large-scale LSM simulations as it is the integral of all hydrological processes occurring in a catchment (Fekete et al., 2012). Moreover, observed river discharge time series are globally available for a large number of catchments (Hannah et al., 2011). LSMs, which were originally developed to provide lower boundary
conditions and vertical fluxes for AGCMs, can be coupled with river routing models that simulate horizontal channel water movement from headwaters to oceans (Ducharne et al., 2003; Pappenberger et al., 2010; Decharme et al., 2010). Anthropic alterations of the natural hydrological regime (e.g. reservoir regulation, abstractions, transfers and man-made drainage



systems) pose challenges for achieving realistic LSM river discharge simulations (Li et al., 2013). While allowing comparisons with the outcome of simulated hydrological processes over a catchment, river discharge does not provide spatially distributed information about model performance at scales finer than the catchment size.

Remote sensing retrievals of the land surface constitute the ideal benchmarks for spatially distributed evaluations of LSMs

(Overgaard et al., 2006), as they provide information integrated over areas whose size is compatible with model resolution. Moreover, there is a growing availability of remote sensing products that are highly relevant for monitoring land-atmosphere interactions. Among those, retrievals of surface soil moisture (SSM) and leaf area index (LAI), which is the total photosynthetically active leaf area per ground unit area, are of particular importance. SSM and LAI are primary controls on the energy, water and carbon fluxes at the land-atmosphere interface. They are both key factors in determining

evapotranspiration and surface albedo. SSM influences infiltration and surface runoff generation, providing the upper boundary conditions for moisture redistribution in unsaturated soils. LAI controls plant phenological development and canopy interception of precipitation.

Remote sensing estimates are affected by uncertainties that may stem from instruments, retrieval algorithms, or external atmospheric and land surface data necessary for the indirect estimation of geophysical variables. Thus, the cross-

comparisons between remote sensing retrievals and LSM simulations provide useful insights for improving both products (see e.g. Brut et al., 2009; Lafont et al., 2012; Albergel et al., 2013a,b; Szczypta et al., 2014; Polcher et al., 2016; Barella-Ortiz et al., 2017). Furthermore, these exercises are useful for the development of land data assimilation systems that aim at integrating independent measurements of key state variables such as SSM and LAI into LSMs (Rodell et al., 2004; Draper et al., 2012; Barbu et al., 2014; Albergel et al., 2017).

Here we present a hydrological evaluation of the impacts of atmospheric forcing uncertainties on LSM simulations. Simulations are carried out from 1979 to 2012 over the Euro-Mediterranean area at 0.5° longitude/latitude resolution. We use the ISBA-A-gs LSM (Noilhan and Planton, 1989; Calvet et al., 1998) coupled with the CTRIP (standing for CNRM TRIP) river routing model (Decharme et al., 2010), an upgraded version of the TRIP model (Oki and Sud, 1997), to simulate the energy and water balances of the land surface as well as plant phenological development. ISBA-A-gs is a $CO_2$-

responsive LSM that simulates vegetation dynamics and provides prognostic estimates of LAI (in the configuration described by Gibelin et al., 2006). It also features a multi-layer diffusion scheme for the soil water and energy balances (Decharme et al., 2013). The latter allows simulating soil moisture over a surface layer whose thickness is compatible with the generally acknowledged penetration depth of L-band microwave radiometers used to remotely sense SSM ($10^{-2}$–$10^{-1}$ m, according to Escorihuela et al., 2010, and Kerr et al., 2010).

To account for forcing uncertainties, LSM simulations are driven by four atmospheric datasets. Three of them are obtained from the ERA-Interim reanalysis (Dee et al., 2011) produced by European Centre for Medium Range Weather Forecasts (ECMWF; Reading, UK). ERA-Interim is the first dataset, while its two derivatives are: a version corrected using the Global Precipitation Climatology Project (GPCP) monthly precipitation data (Huffman et al., 2009); and WATCH Forcing Data ERA-Interim (WFDEI; Weedon et al., 2014) obtained by correcting precipitation, air temperature and downward short-wave





(SW) radiation with monthly data from Climatic Research Unit (CRU; University of East Anglia, Norwich, UK; Harris et al., 2014). These three forcing datasets are characterised by the same large-scale synoptic variability and precipitation occurrences. The fourth dataset is chosen to have a LSM simulation independent from ERA-Interim: PGF (Sheffield et al., 2006) developed by the Terrestrial Hydrology Research Group at Princeton University (Princeton, NJ, USA).

The evaluation provides an overview of how atmospheric forcing uncertainties influence the simulation of surface hydrology. To frame uncertainties and their impacts, model input and output variables are compared to state-of-the-art reference datasets. Forcing data are compared to: E-OBS (van der Schrier et al., 2013) air temperature and precipitation; Global Precipitation Climatology Centre (GPCC; Deutscher Wetterdienst, Offenbach, Germany; Schneider et al., 2014) precipitation; CRU air temperature; NASA/GEWEX Surface Radiation Budget (SRB; Zhang et al., 2015) and Clouds and

the Earth's Radiant Energy System (CERES; Wielicki et al., 1996) for long-wave (LW) and SW downward radiation. Some of these datasets are used to correct biases of atmospheric forcing (see Section 3.1) and thus are not independent references. Simulated LAI and SSM are compared to the remotely sensed Global Inventory Modeling and Mapping Studies (GIMMS; Zhu et al., 2013) and European Space Agency's Water Cycle Multi-mission Observation Strategy and Climate Change Initiative (ESA-CCI; Dorigo et al., 2014) datasets. River discharge simulations are evaluated using the recorded time series

from 99 gauges of the Global Runoff Data Centre (GRDC, 2016) dataset.

The presented assessment involves the atmospheric conditions driving the simulation (forcing variables), the distributed land surface dynamics controlling fluxes at the interface with the atmosphere (LAI and SSM), and the integral of all catchment hydrological processes (river discharge). Moreover, we use long time series of remote sensing retrievals: GIMMS LAI data are available from 1981 to 2011; and ESA-CCI SSM covers the whole simulation period. Several studies evaluated the

impacts of forcing uncertainties on large-scale LSM simulations: Guo et al. (2006) tested the sensitivity of soil moisture simulations forced by several meteorological forcing datasets; Decharme and Douville (2006) and Szczypta et al. (2012) looked at the impact of forcing precipitation errors on river discharge simulations; Materia et al. (2010) analysed the sensitivity of simulated river discharge to changes in the atmospheric forcing data; Nasonova et al. (2011) investigated the impacts of forcing and surface parameter uncertainties on simulated runoff and evapotranspiration; Liu et al. (2016) ran

multi-forcing and multi-model experiments to compare simulated and observed evapotranspiration. To our best knowledge, the extensiveness of the presented assessment exceeds those of previous studies focused on assessing LSM sensitivity to atmospheric forcing uncertainty. The large spectrum of comparisons and the length of the benchmark time series contribute to the completeness of the exercise.

The remainder of the paper is organised as follows. Sect. 2 describes the ISBA-A-gs LSM and CTRIP river routing model.

Sect. 3 presents the atmospheric forcing datasets as well as all data sources used for evaluating the simulations. Sect. 4 outlines the experimental design of simulations and evaluation of model input and output. In Sect. 5 we show the results, which are discussed further in Sect. 6. Finally, Sect. 7 summarises the main findings of this study. The Supplement presents complementary results.



## 2 SURFEX-CTRIP model

To simulate the continental water balance and the related biophysical processes, we use the SURFEX (SURFace EXternalisée) version 7.3 modelling platform (Le Moigne, 2009) coupling the ISBA-A-gs LSM (Calvet et al., 1998) with CTRIP through the OASIS3-MCT coupler (https://verc.enes.org/oasis). Given the atmospheric forcing, ISBA-A-gs

computes the energy, water and carbon balances of spatially distributed independent columns of soil and vegetation. The resulting surface runoff and soil drainage are then routed by CTRIP through the river networks to simulate streamflow.

ISBA-A-gs is a $CO_2$-responsive version of the Interaction between Soil Biosphere Atmosphere (ISBA; Noilhan and Planton, 1989; Noilhan and Mahfouf, 1996) LSM simulating photosynthesis. It is based on the biochemical model of Goudriaan et al. (1985) modified by Jacobs et al. (1996) that describes the relation between photosynthesis and leaf stomatal aperture in the

absence of water scarcity. ISBA-A-gs simulates two types of response to soil moisture stress for both herbaceous and forest vegetation (Calvet, 2000; Calvet et al., 2004): the drought-avoiding response increases water use efficiency at the emergence of the stress; while the drought-tolerant response decreases or keeps the water use efficiency stable throughout the stress period. ISBA parameters depend on soil and vegetation types. Land surface classifications are provided by the ECOCLIMAP-II/Europe database (Faroux et al., 2013) that covers Europe at 1 km resolution and is based on satellite

observations.

We use the following ISBA-A-gs settings:

1. Coupling between above-ground biomass dynamics and soil water balance ("NIT") (Calvet and Soussana, 2001; Gibelin et al., 2006).

2. Multilayer soil diffusion scheme explicitly solving the soil water and heat balance equations ("DF") (Boone et al.,

2000; Decharme et al., 2011; Decharme et al., 2013).

The basic hydrological unit of ISBA is a snow-vegetation-soil composite column. Each model grid cell consists of several homogeneous columns, to account for land surface heterogeneities at a finer spatial scale than the one imposed by the atmospheric forcing. The simulated water balance accounts for: canopy interception of precipitation; throughfall; snow accumulation and melting; infiltration; evapotranspiration; surface runoff; topsoil lateral subsurface flow; and soil drainage.

The snowpack is simulated using a three-layer scheme (Boone and Etchevers, 2001). Soil ice is modelled according to Fuchs et al. (1978). Surface runoff fluxes due to both infiltration- and saturation-excess as well as topsoil lateral subsurface flow are modelled using subgrid approaches (Decharme and Douville, 2006).

Heat and water balances are computed for each column at half-hourly time steps. The fluxes needed for coupling ISBA with other models are then aggregated by grid cell. Thus, daily grid-cell averages of surface runoff and deep soil drainage are

used to force the CTRIP river routing model described in the following section.

CTRIP routes the runoff and drainage flows from ISBA using two reservoirs per model grid cell (Decharme et al., 2010). Deep soil drainage feeds a linear groundwater reservoir, which in turn feeds the surface reservoir with an output flow proportional to its storage through a time constant.



The groundwater reservoir is not meant to represent actual groundwater processes, but to mimic the delay of soil drainage contributions to river discharge. As in Decharme et al. (2012), the groundwater reservoir time constant is fixed to 30 days.

The surface reservoir represents the segment of river channel contained in the grid cell. It is fed by groundwater flow and ISBA surface runoff from its grid cell, as well as by the outflows from the immediately upstream surface reservoirs. Its outflow is proportional to the ratio between channel storage and length through a variable flow velocity, which is given by Manning's formula (Arora and Boer, 1999). The CTRIP river channel parameters (length, width, roughness, and slope) are computed at 0.5° spatial resolution following Decharme et al. (2010), who also provide a detailed description of the model configuration used in this study.

## 3 Data

In this section we describe all datasets used as input and to evaluate model simulations, namely: atmospheric forcing and reference data (Sections 3.1 and 3.2); SSM and LAI remote sensing retrievals (Section 3.3); and river discharge measurements (Section 3.4).

The simulation grid covers the continental surfaces of the Euro-Mediterranean area (25°–75.5°N, 11.5°W–62.5°E) with a homogeneous 0.5° longitude/latitude resolution (Fig. 1).

### 3.1 Atmospheric forcing

ISBA-A-gs simulations were forced by 3-hourly time series of precipitation, downward LW and SW radiation, air temperature and humidity, wind speed, atmospheric pressure, and $CO_2$ concentration. The atmospheric datasets used to force model simulations are briefly described in the following list:

- ERA-I: ERA-Interim reanalysis (Dee et al., 2011) .
- P-ERA: ERA-I (Balsamo et al., 2015) whose precipitation amounts are bias-corrected so that their monthly values match those of GPCP data.
- WFDEI: WATCH Forcing Data ERA-Interim (Weedon et al., 2014). It is obtained from ERA-I by correcting precipitation, air temperature and downward SW radiation with monthly CRU data. Air temperature, pressure and humidity, as well as downward LW radiation are sequentially elevation-corrected. Moreover, air temperature is also corrected using mean monthly diurnal temperature ranges. Downward SW radiation is corrected for cloud cover and aerosol loading. Precipitation amounts are corrected to match CRU monthly averages.
- PGF: Global Meteorological Forcing Dataset for land surface modeling. It is based on the National Centers for Environmental Prediction-National Center for Atmospheric Research (NCEP-NCAR; Boulder, CO, USA; Kalnay et al., 1996) reanalysis. Precipitation and air temperature are bias-corrected so that monthly averages match those of CRU data; as for WFDEI, air temperature, pressure and humidity, as well as downward LW radiation are sequentially elevation-corrected; downward LW radiation is corrected with the NASA/GEWEX Surface Radiation




Budget (SRB) monthly data (Zhang et al., 2013; Zhang et al., 2015); and downward SW radiation is corrected using both SRB and CRU data.

P-ERA and WFDEI inherit large-scale synoptic atmospheric circulation patterns and precipitation occurrences from ERA-I. Therefore, in the following we refer to ERA-I, ERA-P and WFDEI as ERA-based forcing. While WFDEI and PGF were available at 0.5° (latitude/longitude) horizontal resolution, ERA-I and P-ERA were extrapolated from their original 0.75° resolution to 0.5°. All datasets span the experiment period 1979–2012. We used an experimental version of PGF, as the stable version available when simulations were run covers the period 1948–2008 (PGF, 2016). While ERA-based forcing provides rainfall and snowfall, PGF precipitation values were partitioned using an air temperature threshold of 1°C.

The choice of the forcing datasets was motivated by two rationales. First, we wanted to compare the performance of a reanalysis product (ERA-I) with several bias-corrected versions. This type of atmospheric forcing is likely to be affected by biases, which in turn constitute a major source of uncertainty for land surface and hydrological simulations (Berg et al., 2003; Sheffield et al., 2004; Ngo-Duc et al., 2005; Weedon et al., 2011). Second, we included a forcing dataset based on a different reanalysis than ERA-I, i.e. PGF.

## 3.2 Atmospheric reference datasets

Reference datasets were processed for the following atmospheric forcing variables: precipitation, air temperature, and downward LW and SW radiation. To account for uncertainties in state-of-the-art climatological data, we used two datasets per variable.

Precipitation data were obtained from two station-based gridded datasets: the E-OBS version 11.0 product (Haylock et al., 2008; van der Schrier et al., 2013) covering the 1950–2015 period; and merging the 1900-2010 version 6 reanalysis and the 2011-2014 version 4 monitoring products of GPCC.

The air temperature datasets are: E-OBS, and CRU version 3.21 (see Sect. 3.1) spanning the 1901–2012 period.

The chosen reference datasets for downward LW and SW radiation monthly data are: SRB (see Sect. 3.1) versions 3.0 and 3.1 for SW and LW, respectively, covering the 1983–2007 period; and Clouds and the Earth's Radiant Energy System (CERES) version 2.8 from 2000 to 2014 (Wielicki et al., 1996).

E-OBS, GPCC reanalysis, and CRU datasets were available at 0.5° resolution. GPCC monitoring, SRB and CERES products were interpolated from the original 1° to 0.5° resolution via nearest-neighbour estimation.

## 3.3 Remote sensing retrievals

To evaluate simulated SSM and LAI, we used the ESA-CCI 1978–2014 dataset version 02, and the GIMMS 1981–2011 dataset version 1.3. These data sources were chosen since their temporal coverages are comparable to those of the forcing datasets. While the annual missing observation rates of GIMMS LAI are stationary, those of ESA-CCI SSM exhibit a decreasing trend due to the progressive inclusion of different microwave sensors. Despite its non-homogeneous time



coverage, we decided to use ESA-CCI SSM for the whole simulation period (1979–2012) to perform a long term evaluation of LSM output.

### 3.3.1 ESA-CCI surface soil moisture

The ESA-CCI SSM product merges retrievals from several passive and active microwave satellite sensors (Liu et al., 2011; Dorigo et al., 2014). Although the sensing depth depends on soil moisture itself among other factors, microwave-based retrievals are regarded as informative of the moisture in the top few centimetres of soil (Escorihuela et al., 2010).

The source dataset consists of daily estimates (including missing values) spanning the 1978–2014 period at 0.25° horizontal resolution. First, daily maps were interpolated to the 0.5° model grid by averaging only over model pixels for which at least 50% of the possible retrievals were available. Retrievals flagged for insufficient quality were regarded as missing. As topographic relief is known to negatively affect remote sensing estimates of soil moisture (Mätzler and Standley, 2000), we discarded the time series for pixels whose average altitude exceeded 1500 m above sea level. Data on pixels with urban land cover fractions larger than 15% were also discarded, to limit the effects of artificial surfaces. The altitude and urban area thresholds were set according to Draper et al. (2011) and Barbu et al. (2014), who processed ASCAT SSM retrievals for data assimilation exercises with the ISBA LSM.

The average time interval between two processed SSM values at a model pixel is 3.7 days. Splitting the domain at 50° N, retrievals are more frequent in the southern (on average every 3.1 days) than in the northern part (4.5 days). the lower availability at high latitudes is partly explained by winter occurrences of frost and snow cover (Dorigo et al., 2014).

### 3.3.2 GIMMS leaf area index

The GIMMS LAI product is based on an artificial neural network algorithm. The algorithm was trained to map GIMMS normalized difference vegetation index (NDVI) to fraction of absorbed photosynthetically active radiation (FAPAR) and LAI retrieved by the Terra Moderate Resolution Imaging Spectroradiometer (MODIS; Yang et al., 2006). The training was carried out on the overlapping 2000–2009 period. The algorithm was then used to generate LAI and FAPAR fortnightly estimates for the 1981–2011 period at 1/12° spatial resolution (Zhu et al., 2013).

As for ESA-CCI SSM, each retrieval map was interpolated only on those model pixels where missing data were below 50%. The average missing rate in processed data ranges from 34% in summer to 41% in winter, when snow cover hinders the retrievals in mountainous areas and in the northernmost part of the domain.

### 3.4 GRDC river discharge

To evaluate river discharge simulations, we used monthly gauge-based estimates provided by the Global Runoff Data Centre (GRDC, 2016). Ninety-nine GRDC gauges were paired with model pixels. For each gauge, 9 potential matching pixels were considered, namely the pixel containing the gauge and its 8 neighbours. The paired pixel was chosen as the one having the closest upstream drainage area to that of the gauge upstream catchment, provided that it fulfilled the following criteria:



minimum record length of 120 months; minimum upstream catchment area of $2 \cdot 10^4$ km$^2$; maximum absolute difference of 10% between the catchment areas reported by GRDC and those described by the CTRIP river network. These criteria aim at ensuring a meaningful comparison between observed and simulated values. Their enforcement is necessary for coping with the significant distortions in the model representation of the river network that are caused by the coarse spatial resolution.

Model pixels paired with GRDC gauges and the corresponding drained CTRIP river network are illustrated in Fig. 1. Catchment, location, upstream area and observation period of the 35 downstream gauges are reported in Fig. 2, whose rows are indexed with the station numbers mapped in Fig. 1.

## 4 Experimental design

As our objective is the impact assessment of atmospheric forcing uncertainties on land surface hydrology, we performed a
model simulation for each of the four considered forcing datasets (Sect. 3.1). All simulations span the 1979–2012 period, which is the longest overlapping time interval across the forcing datasets.

Model input and output are compared with a number of datasets, which are described in Sect. 3.2 to 3.4. Input values of precipitation, air temperature, as well as downward LW and SW radiation are compared to station- and satellite-based estimates. Among the model outputs, particular attention is dedicated to river discharge, SSM and LAI: river discharge can
be seen as the integral of all catchment hydrological processes; SSM and LAI represent key components of the land surface biophysical state, as they are important controls on water, heat and carbon fluxes at the land-atmosphere interface. While modelled SSM and LAI are compared to remote sensing retrievals, simulated river discharge time series are evaluated with gauge measurements. To have an overview of how the model simulates large scale hydrological processes, average annual cycles of forcing and simulated variables relevant to the water cycle are computed over large catchments.
ISBA-A-gs and CTRIP can produce outputs at half-hourly and daily time intervals, respectively. However, since three of the four forcing datasets are characterised by the same (ERA-I) synoptic variability, we choose to perform our analysis at the monthly time scale. Indeed, we believe that a hydrological evaluation of the uncertainty deriving from the considered forcing datasets at (sub-)daily temporal scale may not add significant further information, because the ERA-based forcings have the same precipitation occurrence processes. Moreover, a daily time scale analysis may be severely affected by the lack of model
representation of anthropic river regulation, which affects most major European rivers.

To compare model input/output variables with corresponding datasets, monthly series are computed using only the time steps where reference data are available. Monthly anomalies are computed from monthly averages: at each grid-cell, the climatological mean and standard deviation are estimated for each calendar month. Then monthly averages are subtracted the climatological mean and divided by the climatological standard deviation to obtain normalised monthly anomalies. Anomaly
statistics measure how well the model simulates departures from the annual cycle, i.e. the inter-annual variability.





### 4.1 Surface soil moisture

The water content of the ISBA top soil layer is compared to the ESA-CCI SSM remote sensing retrievals. We use the top layer, which is 1 cm thick, to take full advantage of the fine vertical soil discretisation of the DF scheme. The soil column is represented with 14 layers over a 12 m depth following Decharme et al. (2013), who simulated soil moisture only over the root zone. Our choice is also based on the conclusions reached by previous studies. Escorihuela et al. (2010) studied the penetration depth of L-band microwave radiometry to estimate SSM: they found that brightness temperature, which is measured directly by the radiometer, was best correlated with soil moisture in the top 2 cm for dry soils and in the top 1 cm for wet soils. Dorigo et al. (2014), who validated ESA-CCI SSM data using ground-based measurements, used the records closest to the surface among those available within the top 10 cm.

To ensure comparability between model output and remotely sensed data, each SSM time series is linearly rescaled to the interval [0, 1] with respect to its minimum and maximum values. The resulting rescaled surface soil moisture (RSSM) can be regarded as a proxy of the topsoil saturation degree (Wagner et al., 1999; Albergel et al., 2012; Parrens et al., 2012; Polcher et al., 2016). This pixel-wise linear rescaling is designed to filter the effects of the discrepancies in the soil properties used as input by the LSM and the remote sensing retrieval algorithm. For instance, different soil composition maps may lead to different estimates of properties such as wilting point or field capacity and, in turn, to differences in soil moisture variability.

### 4.2 Leaf area index

Simulated LAI is compared to GIMMS data, which are available fortnightly. As the exact GIMMS estimate dates are unknown, monthly averages are computed from model values available on the 8[th] and 23[rd] days of each month. This choice is somewhat arbitrary and implies a 1 week uncertainty in the model dates for comparison. Zhu et al. (2013) estimated the root mean squared error of GIMMS LAI to be of the order of 0.5–0.9 $m^2 m^{-2}$. This uncertainty estimate is comparable to the average LAI monthly growth rates of both ISBA-A-gs and GIMMS over the study area (see the averaged LAI annual cycles in Fig. S2 to S5 of the Supplement). Thus, we believe our assumption would cause errors that are not larger than the uncertainties stemming from the GIMMS algorithm.

### 4.3 River discharge scores

Monthly river discharge simulations are compared to GRDC gauge measurements using several statistical scores. Relative errors of simulated mean ($RE_\mu$) and standard deviation ($RE_\sigma$) account for biases and variability errors:

$$RE_\theta = \frac{\theta_{sim}}{\theta_{obs}} - 1 \tag{1}$$

where $\theta$ may be either $\mu$ or $\sigma$ (standing for sample mean or standard deviation), while the subscripts *sim* and *obs* indicate whether the source data is simulated or observed. Correlation coefficients based on monthly time series (MC) quantify how accurately the simulations reproduce the shape and timing of measurements.



$RE_\mu$, $RE_\sigma$ and MC constitute a set of mathematically independent score functions quantifying the goodness of fit of simulated discharge time series. At their ideal values ($RE_\mu$, $RE_\sigma = 0$ and MC = 1), the simulated time series is identical to the measured one. Moreover they account for errors in modelling the first three fundamental hydrological functions of a watershed as identified by Black (1997), Wagener et al. (2007) and Yilmaz et al. (2008): the overall water balance resulting from the partition of precipitation between evapotranspiration and infiltration plus direct runoff ($RE_\mu$); the soil water storage dynamics distributing the excess precipitation among faster and slower runoff components ($RE_\sigma$); and the water release and routing processes determining timing and shape of the hydrograph (MC).

To summarise the information of $RE_\mu$, $RE_\sigma$ and MC with a single aggregate score, we compute the Kling-Gupta efficiency (KGE; Gupta et al., 2009):

$$\text{KGE} = 1 - \sqrt{RE_\mu^2 + RE_\sigma^2 + (1 - \text{MC})^2} \qquad (2)$$

which is the Euclidean distance from the ideal point in the [$RE_\mu$, $RE_\sigma$, MC] score space. We prefer KGE to other widely used residual-based summary statistics such as the Nash-Sutcliffe efficiency (NSE; Nash and Sutcliffe, 1970) or the related mean squared error (MSE), to prevent the following drawbacks identified by Gupta et al. (2009): in catchments with high discharge variability, NSE and MSE overrate model simulations with large biases; and if MC is smaller than 1 (always in real cases), simulations underestimating discharge variability are overrated. While NSE = 0 means that the model is a worse predictor than the observed average, negative KGE values have no defined interpretation. This advantage of NSE, however, is significant in the domain of poor model performance.

To evaluate the model ability to capture the inter-annual variability at the monthly scale, we compute correlations between anomaly time series (AC). AC are based on normalised departures from the averaged discharge annual cycle. Thus they allow filtering out the correlation induced by the climatological seasonality, which contributes significantly to MC.

### 4.4 Catchment averaged annual cycles

We compute catchment averaged annual cycles for the following forcing variables: downward LW and SW radiation, air temperature at 2 m, wind speed, relative air humidity, and total precipitation. Of the simulated land surface states and fluxes, we show: river discharge, evapotranspiration, LAI, and RSSM. To be consistent with river discharge, which is the integral of upstream hydrological processes, annual cycles are computed using only the time steps when river discharge measurements are available.

### 5 Results

The evaluation of model simulations against the datasets described in Sect. 3 is reported in the following. Catchment averaged simulated annual cycles of hydrologically relevant variables are described in the Supplement for four river basins (Danube, Rhone, Ebro and Po).





## 5.1 Surface soil moisture

We compare the simulated water content of the top soil layer of the ISBA model with the ESA-CCI SSM remote sensing retrievals (Sect. 3.3.1). To ensure comparability, each SSM time series is linearly rescaled with respect to its minimum and maximum, as motivated in Sect. 4.1.

Global correlation values are computed using the entire datasets and are reported in the upper-left corners of the maps in Fig. 3. As expected, similar scores are obtained for the different simulations, among which the PGF-forced SSM is slightly less correlated with ESA-CCI data. Looking at this small difference, we must recall that the atmospheric synoptic variability of PGF is different from that of ERA-based forcing. ERA-I, P-ERA and WFDEI are characterised by the same precipitation occurrences, while the intensities of individual precipitation events may differ according to the applied bias-correction (Sect.

10  3.1).

Temporal correlation maps between simulated and remotely sensed SSM are computed for both monthly absolute and anomaly time series (Fig. 3). Pixel-wise temporal correlation is not affected by the rescaling. Similar correlation patterns emerge across the simulations. Monthly time series are most correlated (between 0.5 and 1) in southern and western Europe, northern Africa, Middle East, and the portion of central Asia included in the domain. Smaller positive correlations (below

0.5) characterise large areas of central and eastern Europe, the Caucasus, and south-eastern Russia. Null or negative correlations are found in Scandinavia, eastern Baltic and northern Russian regions, as well in mountainous areas such as the Alps and the Carpathians. Possible causes for these low values may be vegetation density, topography and soil frost occurrences, all of which are known to negatively affect SSM retrievals. Despite generally smaller values, temporal correlations of anomalies exhibit map patterns that are similar to those described for monthly values.

Fig. 3 takes the ERA-I correlation as a reference to show the departures of the other forcing datasets. Consistently with the above-mentioned similarities between correlation maps, absolute differences are small: they are below 0.1 for at least 90% (monthly averages) and 74% (anomalies) of the mapped grid-cells. Over 94% of P-ERA monthly and anomaly correlations are within a 0.05 range of ERA-I values, with small positive differences scattered in the eastern part of the domain. The same is true for 81% (70%) and 66% (44%) of, respectively, WFDEI and PGF monthly values (anomalies). WFDEI negative

departures are mostly located in northern Africa. PGF yields the largest differences, particularly those of negative sign.

Correlation coefficients computed separately for each calendar month (Fig. 4) report the seasonal variations in the agreement between simulated and remotely sensed RSSM. The correlation annual cycles of monthly absolute values reach their peaks from June to September and their minima in November-December and March-April, indicating that RSSM maps are more consistent during the driest months. All monthly anomaly correlations have their maxima in September and their minima

from December to March. Thus the agreement between deviations from RSSM climatologies tend to steadily increase through summer, peaking when soils are driest. While PGF correlations are systematically the smallest for both monthly absolute and anomaly values, the differences are not dramatic and their cycle shapes are similar to those of ERA-based data.





The similarities of (R)SSM correlation statistics across the simulations suggest that the model reacts robustly to forcing uncertainty. However, it may also indicate that the discrepancies found between modelled and remotely sensed (R)SSM could be caused by the LSM representations of physical processes and the remote sensing retrieval algorithms. The correlation annual cycles (Fig. 4) point in the same direction. Indeed, correlations are largest in summer when SSM

variability is minimum due to dry conditions. Instead they are smallest in winter, when SSM variability is maximum due to precipitation and thus the description of SSM dynamics becomes more important.

## 5.2 Leaf area index

Simulated phenology is compared to the satellite based GIMMS LAI product (Sect. 3.3.2).

Temporal correlations of monthly averages are large (above 0.7) over more than 75% of the pixels where data are available,

for all forcing datasets (Fig. 5). The largest values are mostly in the central and eastern parts of the domain. Very small or even negative correlations are scattered across the southern part. Irrigation, which is not simulated by ISBA-CTRIP, may be partially accountable for these discrepancies in some areas, as for instance the Nile delta and Mesopotamia.

Anomalies are systematically less temporally correlated than monthly values in all simulations. Over 90% of pixels with data availability score values below 0.5. However, values below 0.1 are less than 3%, so the areas with null or negative

correlations are very limited. Consistently with temporal correlation maps, global correlation coefficients (upper left corners of maps in Fig. 5) are large for monthly averages (0.79–0.81) and significantly smaller for anomalies (0.32–0.34).

Fig. 5 maps ERA-I temporal correlation coefficients and the differences yielded by the other simulations, as done for SSM (Fig. 3). Discrepancies between correlation maps of different simulations are generally small. Over 94% and 89% of the absolute differences with ERA-I correlations are smaller than 0.1 for monthly values and anomalies, respectively. P-ERA

yields the most similar maps to ERA-I: more than 95% and 83% of monthly and anomaly correlation differences are below 0.05. This is expected, as P-ERA is identical to ERA-I except for precipitation amounts. Consistently, WFDEI and PGF, which feature progressively increasing divergences from ERA-I forcing, yield slightly larger correlation differences. These are below 0.05 for 82% (69%) and 81% (61%) for monthly (anomaly) correlations of WFDEI and PGF. While differences from ERA-I correlations are prevalently positive for P-ERA, their signs are more mixed for the other two forcing datasets.

The largest clearly distinguishable difference signals, which are yielded by WFDEI and PGF monthly correlations, are located along the Caucasus and Pontic mountain ranges.

LAI correlation statistics show a general agreement across forcing datasets. This is also true for monthly correlation coefficients (Fig. 6), which follow very distinguishable seasonal patterns. Correlations between monthly averages peak in July-August (0.85), when LAI reaches its maximum. Instead, anomaly correlation are largest (0.50–0.55) in April–May,

when the phenological development rate is likely to be maximum. Both correlation annual cycles are lowest in winter, when the ISBA-A-gs dynamic vegetation model sets LAI of several vegetation types to prescribed values until the next leaf onset occurs triggering a new growing season. This does not apply to evergreen trees and winter crops. The prescribed minimum





LAI values may be partly accountable for the small winter correlations. They may have a particularly negative impact on anomalies by hindering any winter inter-annual variability.

All correlation statistics are very stable across the forcing datasets. This is similar to what observed for SSM, with two exceptions: PGF statistics are more aligned with the other forcing datasets; and correlations for monthly averages are generally larger. Thus, compared to SSM, LAI correlation statistics are less sensitive to forcing uncertainty. This is not surprising as the impact of precipitation uncertainty is larger on SSM than on LAI. Indeed, LAI dynamics are strongly influenced by the periodicity of plant development and by forcing variables that are less uncertain than precipitation, such as air temperature and SW radiation. In other words, the LAI degrees of freedom are reduced compared to SSM. Furthermore, LAI is less sensitive than SSM to short-term uncertainties in the atmospheric forcing, due to its larger characteristic time scale.

## 5.3 River discharge scores

Simulated monthly river discharge time series are evaluated using the GRDC dataset (Sect. 3.4). We use the statistical scores defined in Sect. 4.3: $RE_\mu$, $RE_\sigma$, MC, AC, and KGE. Score values are summarized in exceedance frequency plots (Fig. 7) for all 99 selected GRDC gauges. Moreover, they are reported in detail for the 35 most downstream gauges (Tables 1 and 2).

To facilitate graphical display and interpretation, Fig. 7 plots the absolute values of $RE_\mu$ and $RE_\sigma$. The abscissae of $|RE_\mu|$ and $|RE_\sigma|$ are inverted, so that all exceedance frequency plots can be read in the same way: the higher the curve, the better the performance.

The simulations yield similar MC and AC summary curves (Fig. 7). Approximately 60% of the gauges have MC and AC values above 0.6. The WFDEI and P-ERA simulations slightly outperform the others in terms of MC and AC, respectively, but differences are small. The inter-forcing spread becomes larger if we look at relative errors. In terms of $RE_\mu$, ERA-I performs best followed by PGF, WFDEI and P-ERA. While $|RE_\mu|$ is below 0.2 for over 70% of ERA-I-forced time series, the same happens for less than 50% of P-ERA series. The spread is even wider for $|RE_\sigma|$. The PGF simulation dominates almost the entire frequency spectrum, followed by WFDEI, ERA-I and P-ERA. More than 60% of PGF and less than 25% of P-ERA series have $|RE_\sigma|$ below 0.4. The $|RE_\mu|$ and $|RE_\sigma|$ exceedance frequency curves of P-ERA are almost systematically dominated by those of the other simulations. The relative errors have a strong impact on the summary curves of the aggregate score KGE, according to which the P-ERA simulation is dominated at almost all frequencies. Instead, a clear KGE ranking cannot be established among the other three simulations.

The spatial patterns of river discharge scores are mapped and discussed in the Supplement (Fig. S1). The main findings are summarised here: the similarities between correlation maps indicate that the timing and shape of simulated monthly discharge are relatively insensitive to forcing uncertainty; relative error maps are also similar, but larger discrepancies emerge across simulations; in particular, P-ERA scores the largest $|RE_\sigma|$ over several catchments (consistently with Fig. 7); KGE spatial patterns are largely influenced by those of relative errors, in particular $|RE_\sigma|$.





River discharge scores computed at gauges having downstream/upstream connections may carry redundant information. Thus simulations performing systematically better/worse than others in multi-gauged catchments may be overrated/underrated, as such catchments would be virtually assigned larger weights when pooling the scores for comparison (Fig. 7 and S1). To avoid redundancy, we look at river discharge scores at the most downstream gauges of each catchment

(Tables 1 and 2). Discharge series at these gauges are the closest estimates of surface water flowing into the oceans. Thus, detailed assessments of discharge simulations at these river cross-sections are of crucial importance, both to validate the simulated water balance of large catchments and for the integration with coastal water and ocean models.

All simulations overestimate the standard deviation ($RE_\sigma > 0$) of discharge time series at the majority of downstream gauges (Table 1). Systematic underestimation occurs only for Severnaya Dvina, Mezen and Torneaelven rivers, which drain sub-

Arctic catchments. At 30 of 35 downstream gauges the P-ERA simulation overestimates average discharge ($RE_\mu > 0$), while for other simulations negative and positive biases are fairly balanced. In terms of $RE_\mu$ and $RE_\sigma$, P-ERA is inferior to other simulations at 20 and 27 downstream gauges, respectively.

MC is above 0.5 at over half of the downstream gauges for all simulations (Table 1). The same applies to AC, whose values are not systematically inferior to MC (Table 2): for simulated time series with MC above 0.5, AC tends to be slightly smaller

than MC; the opposite happens for MC below 0.5. Catchments with larger MC and AC values are located in central and western Europe (Danube, Rhone, Tejo, Weser, Meuse), as also shown in Fig. S1. Relatively low scores characterise some northern and eastern European catchments (Volga, Don, Neva, Ural, Vuoksi, Kovda, Luleaelven). P-ERA yields the best MC values for more than a third of the gauges. Both ERA-I and PGF, which perform better than the other simulations in terms of $RE_\mu$ and $RE_\sigma$, produce the worst MC values for over a third of the gauges. Similar counts are observed for AC as well.

The P-ERA simulation yields the worst KGE at 25 downstream gauges (Table 2). Of these gauges, for 20 it also delivers the worst $RE_\mu$ and $RE_\sigma$ scores; for 8, it yields the best MC; and the worst MC only for 3. This is in agreement with the large influence of $RE_\mu$ and $RE_\sigma$ on KGE already found when discussing Fig. 7 and S1. Moreover, this is caused by forcing uncertainty having the largest impacts on the mean and standard deviation of simulated discharge. Instead, shape, timing and inter-annual variability (through anomaly correlation) are less sensitive to forcing uncertainty.

**5.4 Catchment averaged annual cycles**

To have an overview of how atmospheric forcing uncertainty affects simulated hydrology at catchment scale, we compare the average annual cycles of forcing and LSM output variables over four catchments: Danube, Rhone, Ebro and Po. These catchments were chosen as they contribute to the water balance of the Mediterranean sea, even if indirectly as the Danube; and because the model reproduces fairly well discharge measurements at their downstream GRDC gauges (Tables 1 and 2).

The computed catchment averaged annual cycles are shown and described in detail in Sect. S2 of the Supplement (Fig. S2 to S5). The main findings are summarised in the following paragraphs.

Precipitation features the most different annual cycles across atmospheric datasets. While forcing values are often larger than measurement-based averages, the seasonal distributions are generally in good agreement within each catchment.



Simulated annual cycles of RSSM are close. The pixel-wise rescaling (Sect. 4.1) is partly accountable for the similarities, which are particularly strong among ERA-based simulations. Simulated cycles are in phase with ESA-CCI RSSM, but their amplitudes are generally overestimated. This may be due to structural differences between modelled and remotely sensed SSM that are beyond first-order soil property discrepancies, which should have been filtered out by the rescaling.

Simulated LAI cycles are relatively insensitive to forcing uncertainty. In the Danube, Rhone and Po catchments, they are in good agreement with GIMMS LAI when phenological development is fastest, but they overestimate annual maxima and during senescence. In the Ebro catchment, simulated cycles overestimate the remote sensing-based one.

At most downstream gauges, shape and timing of observed discharge cycles are well reproduced. Simulated cycle amplitudes are generally larger than for measurements: while summer-autumn discharge is underestimated, winter and spring values are overestimated and the inter-simulation spread is maximum, consistently with the precipitation cycles.

## 6 Discussion

To evaluate how atmospheric forcing uncertainties affect the simulation of land surface hydrological processes, we compared model input and output with several independent datasets. In the following subsections, we discuss the main results and we draw some recommendations for future research.

### 6.1 Atmospheric forcing

Precipitation is the most uncertain forcing variable (Fig. S2 to S5). Conversely, the least uncertain forcing variable is air temperature, followed by SW and LW radiation. Both P-ERA and WFDEI forcing datasets are derived from ERA-I. While P-ERA differs from ERA-I only because of monthly precipitation amounts, WFDEI also features differences in SW radiation, air temperature and relative humidity. PGF is independent from ERA-based datasets. In addition to precipitation, also wind speed and relative air humidity show remarkable discrepancies between PGF and the other datasets.

### 6.2 Surface soil moisture

The similarities in correlation statistics across simulations suggest that modelled SSM is rather insensitive to forcing uncertainty (Fig. 3 and 4). It also suggests, however, that some causes of the discrepancies between modelled and remotely sensed SSM may be rooted in the representations of the physical processes underlying LSMs and remote sensing retrieval algorithms. Advancing the understanding of these discrepancies is crucial for assimilating remotely sensed SSM into LSMs (Reichle et al., 2008; Draper et al., 2012; Carrera et al., 2015; Barbu et al., 2014; Fairbairn et al., 2015). In light of the presented results, we suggest the following research directions: joint assessments of forcing and model uncertainty (Entin et al., 1999; Guo et al., 2006; Materia et al., 2010; Nasonova et al., 2011); extension of LSM-satellite comparisons to land surface state variables and fluxes driving model simulation or remote sensing retrieval of SSM, e.g. soil brightness



temperature (Parrens et al., 2014; Muñoz-Sabater, 2015; Barella-Ortiz et al., 2017) or radar backscattering coefficient (Stoffelen et al., 2017).

The SSM statistics yielded by PGF show a slight departure from the range of ERA-based simulations. As PGF is the only forcing dataset not characterised by ERA-I synoptic variability, this result raises questions about the impact of precipitation occurrences on SSM correlations compared to other forcing variables. Indeed, SSM is likely to be very sensitive to precipitation occurrences, as the modelled/observed surface soil layer is very thin (less than 10 cm). Following investigation steps in this regard should be carried out at finer temporal scales than monthly, at which higher frequency variability is filtered out by the time averaging. For instance, daily correlations would be more informative about the consistency of forcing precipitation events. Thus one could test SSM sensitivity to precipitation occurrences compared to amounts. For this, LSMs should be forced by several atmospheric datasets with independent precipitation occurrences. As in this study 3 of 4 simulations are forced by the same precipitation occurrences, we analysed the output at the monthly time scale.

The ESA-CCI RSSM annual cycles display smaller amplitudes than the simulations, whose cycles are very close. These observations are valid not only for the catchments analysed in Sect. S2 (Fig. S2 to S5), but for all the areas drained by the chosen 35 downstream GRDC gauges (not shown). We may draw a connection between this result and the spectral analysis performed by Polcher et al. (2016), who compared SSM simulated by the ORCHIDEE LSM (Krinner et al., 2005) with SMOS retrievals (Kerr et al., 2010) in the Iberian Peninsula. They found the power ratio of low (seasonal to annual) to high (daily to monthly) frequencies in the SSM signal to be significantly smaller for SMOS than for the LSM. To validate this hypothesis, further large-scale analyses of SSM signals may be needed, also involving several models and forcing datasets to account for as many uncertainty sources as possible.

The rescaling (Sect. 4.1) ensuring comparability between modelled and remotely sensed SSM may be an additional source of uncertainty. For future work, we recommend testing more robust rescaling techniques. For example, one could use physically meaningful parameters such as wilting point and field capacity instead of the time series extreme values, as done here.

### 6.3 Leaf area index

Compared to SSM, simulated monthly LAI is generally more correlated with the remotely sensed estimates; and LAI correlation statistics are more insensitive to forcing uncertainty (Fig. 5 and 6). These differences may be due to the larger impact of precipitation uncertainty on SSM than on LAI. Simulated LAI is relatively insensitive to atmospheric forcing uncertainties, suggesting that phenological dynamics are reproduced robustly by the model.

The correlation between simulated and remotely sensed LAI is largest during the maximum phenological development phase (Fig. 6). This is an encouraging result for the simulation of crop yield and, in general, of the primary production of land surface ecosystems. LAI correlations reach their minima in winter. As ISBA-A-gs prescribes a minimum LAI value for non-growing seasons, future research could evaluate the impact of this assumption on winter correlations and inter-annual variability by means of tailored sensitivity analyses.



Simulated catchment averaged annual cycles of LAI feature systematic overestimations at the maximum phenological development and during senescence (Fig. S2 to S5). This is consistent with the findings of Zhu et al. (2013), who compared the GIMMS LAI annual cycles with an ensemble of 18 Earth system models (see Sect. 4 and Fig. 7 of the cited article). They found GIMMS LAI to be generally below the models mean by approximately the ensemble standard deviation. Moreover,

they pose the question of whether "dynamic vegetation models overestimate carbon fixation and/or allocation of biomass to leaves".

Validation studies on the biomass production simulated by ISBA-A-gs have been performed over agricultural sites in France (Calvet et al., 2012; Canal et al., 2014; Dewaele et al., 2017). Further investigation may be carried out by extending the spatial scale of the validation (see e.g. Smith et al., 2010a,b). Similarly, irrigation modelling and its impacts on simulated

agricultural production need to be validated at large spatial scales: Garrigues et al. (2015) evaluated how ISBA-A-gs simulated evapotranspiration over an irrigated Mediterranean agricultural site. Dynamic vegetation modelling and realistic representations of irrigation are crucial to predict crop yield and assess the sustainability of crop water management (Jägermeyr et al., 2016).

## 6.4 River discharge

Simulated monthly river discharge are generally well correlated with GRDC estimates. Anomaly correlations are comparable to those obtained for the monthly time series. Both MC and AC have similar distributions and spatial patterns across the simulations, indicating that the shape and timing of simulated discharge time series are relatively insensitive to forcing uncertainty (Fig. 7 and S1). The performance spread between simulations becomes larger when we consider $RE_\mu$ or $RE_\sigma$. The P-ERA forced simulation produces the largest relative errors at the majority of the gauges, while ERA-I and PGF yield the

best overall results in terms of $|RE_\mu|$ and $|RE_\sigma|$ respectively (Fig. 7 and Table 1).

Positive and negative biases are well balanced for all simulations except for P-ERA that overestimates average discharge at most downstream gauges (Table 1). In contrast, all simulations tend to overestimate the standard deviation of the measured time series. This overestimation is in agreement with the generally larger amplitude of simulated discharge annual cycles compared to the measurements (Fig. S2 to S4). These results suggest that forcing uncertainty affects more the mean and

standard deviation rather than the timing, shape and inter-annual variability of simulated monthly discharge.

The seasonal distributions of measured and simulated discharge are often different. In particular, simulations tend to underestimate summer-autumn discharge that correspond to the low flow conditions in the study area. Furthermore, in several catchments simulated winter-spring discharge peaks are significantly larger than the corresponding reference data. These systematic errors, which are found in all simulations, may be partly due the lack of model representations of physical

processes and anthropic alterations affecting the streamflow regime.

In the past few years, new components have been tested to provide the ISBA-CTRIP modelling suite with a progressively more complete representation of hydrological processes. Vergnes and Decharme (2012) implemented a simple groundwater scheme coupled to the river channel in each model grid cell allowing bidirectional water exchanges through the riverbed.





Vergnes et al. (2014) introduced the representation of capillary water rise from the groundwater reservoir, which provides the lower boundary condition for the moisture redistribution in the unsaturated soil column. To try to improve the modelled discharge response to rainfall events, the current Richards equations representation of water movement in unsaturated soil might be complemented with other parametrisations as, for instance, preferential flow in macro-pores (Beven and Germann,

5   2013).

Regulation of lakes and artificial reservoirs has a large impact on river discharge (Biemans et al., 2011; Zhou et al., 2016). Its representation in LSMs and hydrological models at global scale has proven to be beneficial to river discharge and irrigation modelling (Hanasaki et al., 2006; Pokhrel et al., 2012). Moreover it allows simulating the impacts of anthropic regulation on water-related activities. Further work should be carried out to embed reservoir regulation modules in LSMs.

River discharge is the integral of all upstream hydrological processes. Land water storage is a fundamental state variable for determining the catchment response to meteorological forcing. While large-scale direct observations are not available, the Gravity Recovery and Climate Experiment (GRACE) satellite mission provides land water storage variation estimates derived from measurements of the terrestrial gravity field variations (Swenson et al., 2003). Several studies have compared LSM simulated land water storage to GRACE estimates (see e.g. Alkama et al., 2010; Becker et al., 2011; Grippa et al.,

2011; Vergnes and Decharme, 2012). If integrated with the evaluation of other components of the land water cycle, these comparisons could help back-tracking the error sources of LSM simulations.

## 7 Conclusions

To assess the hydrological impacts of atmospheric forcing uncertainties, we compared land surface model (LSM) simulations forced by several meteorological datasets against station-based and remote sensing estimates. Simulations were

run for the 1979–2012 period over the Euro-Mediterranean area, which is widely acknowledged for its drought vulnerability especially under climate change scenarios.

We used the ISBA-CTRIP modelling platform, which simulates dynamic plant above-ground biomass and a multi-layer diffusion scheme for the unsaturated soil water and energy balances, together with river discharge, a non-prognostic linear groundwater reservoir, and a non-linear surface reservoir simulating variable-velocity river channel flow.

The simulations were forced using four atmospheric datasets and the evaluation of model outputs focused on surface soil moisture (SSM), leaf area index (LAI), and river discharge.

Simulated and remotely sensed annual cycles of SSM are generally in phase, but simulated amplitudes are systematically larger despite the rescaling. The comparisons indicate that SSM simulations are relatively insensitive to forcing uncertainty. At the same time, the differences between simulated and remotely sensed SSM may be due to structural inconsistencies

between the assumptions used by the LSM and the retrieval algorithm.

LAI temporal correlation maps as well as global correlation coefficients show a good agreement between modelled and observed values for monthly averages. Systematically smaller values are obtained for monthly anomalies. Simulated LAI is





most correlated with remotely sensed data during the typical seasons of maximum phenological development (spring and summer). The lower winter correlations may be negatively affected by the minimum LAI values prescribed by ISBA-A-gs during non-growing seasons. In general, LAI correlation statistics are at least as insensitive to forcing uncertainty as those of SSM.

Precipitation is the most uncertain forcing variable. Nevertheless, the timings of precipitation annual cycles are generally similar within each catchment. The least uncertain forcing variables are air temperature, LW and SW radiation.

The annual cycles of simulated discharge have generally larger amplitudes compared to GRDC records. This translates into almost systematic overestimation of winter-spring high flows and underestimation of summer-autumn low flows. Positive and negative errors in long term average discharge are fairly balanced for all simulations except for P-ERA, which yields

positive biases at most gauges. P-ERA also produces the largest standard deviation errors at the majority of the gauges, while ERA-I and PGF yield the best overall results in terms of bias and standard deviation error, respectively. The spread between simulations is significantly reduced when considering correlations of monthly discharge and anomalies. Moreover, anomaly correlations are not inferior to those of raw monthly values, thus highlighting the model ability to reproduce the inter-annual variability of river discharge. The impacts of forcing uncertainty are larger on the mean and standard deviation rather than

the timing, shape and inter-annual variability of simulated discharge. Moreover, the errors in the seasonal distribution may be due to the lack of model representations of physical processes and anthropic alterations that affect streamflow.

Several results point at structural discrepancies between simulations and reference datasets that seem not to be addressable to forcing uncertainty. Therefore, we identify several research directions to improve LSMs and to foster the assimilation of remote sensing retrievals. The differences between simulated and remotely sensed SSM may be further investigated by

extensively comparing simulated and retrieved land surface variables driving SSM dynamics, such as soil brightness temperature or radar backscattering coefficient (see e.g. Parrens et al., 2014; Stoffelen et al., 2017). To improve the simulation of river discharge and to assess the impacts of water abstractions, LSMs may progressively integrate groundwater modelling (see e.g. Vergnes et al., 2014) and lake and reservoir regulation (see e.g. Hanasaki et al., 2006; Pokhrel et al., 2012). Due its relevance for crop yield and water demand prediction, large-scale irrigation schemes should be embedded into

LSMs (Jägermeyr et al., 2016). Moreover, simulated plant biomass growth should be validated at large spatial scales using agricultural statistics (see e.g. Smith et al., 2010a,b). In general, to better identify the sources of uncertainty in LSM simulations, forcing and model uncertainties may be assessed jointly via extensive multi-forcing and -model experiments (see e.g. Nasonova et al., 2011).

## Acknowledgements

We are thankful to Emanuel Dutra and Gianpaolo Balsamo (European Centre for Medium-Range Weather Forecasts, ECMWF, Reading, UK) for providing the ERA-Interim atmospheric forcing dataset as well as the GPCP-corrected version. The work of Emiliano Gelati was supported by the French REMEMBER project (ANR 2012 SOC&ENV 001) within the





HYMEX initiative. The work of Marie Minvielle and of David Fairbairn was supported by the following European FP7 projects: eartH2Observe (grant agreement 603608) and ImagineS (grant agreement 311766), respectively. Graham P. Weedon was supported by the Joint DECC and Defra Integrated Climate Program - DECC/Defra (GA01101).

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



**Table 1.** River discharge scores (REμ, REσ, MC) computed at downstream GRDC gauges. For each gauge and score pair, the best/worst simulation is highlighted in bold black/grey.

| | REμ | | | | REσ | | | | MC | | | |
|---|---|---|---|---|---|---|---|---|---|---|---|---|
| | ERA-I | P-ERA | WFDEI | PGF | ERA-I | P-ERA | WFDEI | PGF | ERA-I | P-ERA | WFDEI | PGF |
| Volga | **0.08** | 0.46 | 0.11 | 0.19 | 1.19 | 1.93 | 1.62 | **1.17** | 0.14 | 0.18 | **0.21** | 0.16 |
| Danube | **0.03** | 0.19 | 0.05 | 0.04 | 0.52 | 0.88 | 0.41 | **0.27** | 0.77 | 0.73 | 0.86 | **0.87** |
| Dnepr | 0.69 | 1.45 | **0.53** | 0.79 | **2.42** | 4.39 | 2.61 | 2.65 | 0.33 | **0.35** | 0.32 | 0.33 |
| Don | 1.24 | 3.24 | 1.14 | **0.99** | 5.75 | 10.81 | 6.48 | **4.99** | 0.30 | 0.33 | 0.35 | **0.40** |
| Severnaya Dvina | -0.11 | 0.05 | -0.18 | **0.03** | -0.47 | -0.37 | -0.40 | **-0.29** | 0.03 | 0.10 | **0.15** | -0.09 |
| Neva | **-0.05** | 0.32 | -0.09 | 0.13 | **1.49** | 2.47 | 1.54 | 1.61 | -0.41 | -0.42 | -0.38 | **-0.32** |
| Vistula | -0.05 | **0.00** | -0.24 | -0.12 | 0.75 | 1.00 | 0.39 | **0.36** | 0.64 | 0.62 | 0.65 | **0.69** |
| Ural | 1.99 | 2.91 | 1.44 | **0.52** | 4.24 | 5.94 | 3.69 | **1.81** | 0.27 | 0.29 | 0.33 | **0.36** |
| Rhine | **0.01** | 0.24 | 0.19 | 0.09 | 0.40 | 0.81 | 0.43 | **0.34** | 0.79 | **0.82** | 0.54 | 0.61 |
| Elbe | **-0.03** | 0.55 | -0.08 | -0.05 | 0.48 | 1.50 | 0.35 | **0.26** | 0.76 | 0.77 | **0.81** | 0.76 |
| Oder | -0.21 | **0.10** | -0.36 | -0.27 | 0.71 | 1.45 | 0.46 | **0.44** | 0.62 | **0.63** | 0.60 | 0.60 |
| Rhone | -0.05 | 0.12 | 0.02 | **0.00** | 0.43 | 0.66 | **0.26** | 0.30 | 0.82 | **0.85** | 0.76 | 0.76 |
| Ebro | -0.21 | 0.28 | **0.04** | 0.09 | **-0.03** | 0.58 | 0.44 | 0.40 | 0.78 | **0.84** | 0.79 | 0.78 |
| Nemunas | -0.10 | 0.31 | -0.23 | **0.01** | 0.75 | 1.67 | 0.76 | 0.77 | 0.65 | 0.73 | 0.69 | **0.81** |
| Po | -0.36 | -0.17 | **-0.10** | -0.13 | -0.14 | 0.12 | -0.03 | **0.02** | 0.71 | **0.78** | 0.74 | 0.74 |
| Tejo | **-0.09** | 0.31 | -0.22 | -0.26 | **0.07** | 0.51 | -0.12 | -0.20 | 0.87 | **0.90** | 0.86 | 0.84 |
| Daugava | **-0.02** | 0.44 | -0.08 | 0.11 | -0.07 | 0.41 | **0.04** | -0.06 | 0.55 | 0.62 | 0.56 | **0.64** |
| Duero | **0.00** | 0.61 | 0.12 | 0.12 | **0.14** | 0.82 | 0.31 | 0.23 | 0.87 | **0.90** | 0.88 | 0.85 |
| Vuoksi | 0.10 | 0.50 | -0.05 | **-0.03** | 1.92 | 2.85 | **1.68** | 1.70 | 0.13 | 0.14 | 0.15 | **0.15** |
| Guadiana | 0.64 | 1.58 | 0.65 | **0.58** | 0.18 | 0.74 | 0.15 | **0.03** | 0.78 | **0.83** | 0.79 | 0.76 |
| Mezen | -0.21 | **-0.07** | -0.25 | -0.15 | -0.50 | -0.40 | -0.41 | **-0.33** | 0.31 | 0.42 | **0.44** | 0.10 |
| Narva | -0.09 | 0.30 | -0.15 | **0.00** | **1.36** | 2.48 | 1.65 | 1.39 | 0.29 | **0.37** | 0.31 | 0.34 |
| Kemijoki | **0.00** | 0.19 | -0.24 | -0.17 | -0.12 | **0.09** | -0.21 | -0.26 | 0.58 | **0.60** | 0.54 | 0.31 |
| Guadalquivir | **0.47** | 1.73 | 0.60 | 0.52 | 0.25 | 1.22 | 0.38 | **0.23** | 0.80 | 0.86 | **0.87** | 0.82 |
| Goeta | -0.06 | 0.06 | 0.02 | **0.02** | **0.49** | 0.77 | 0.74 | 0.60 | **0.64** | 0.63 | 0.61 | 0.56 |
| Glama | 0.08 | 0.10 | **-0.02** | -0.21 | 0.42 | 0.34 | 0.35 | **-0.14** | 0.62 | 0.63 | **0.77** | 0.47 |
| Weser | -0.31 | 0.25 | -0.28 | **-0.22** | -0.03 | 0.79 | 0.02 | **-0.02** | 0.85 | **0.89** | 0.87 | 0.82 |
| Kymijoki | 0.20 | 0.55 | -0.01 | **0.01** | 1.63 | 2.48 | 1.49 | **1.31** | **0.33** | 0.30 | 0.32 | 0.32 |
| Torneaelven | **-0.01** | -0.04 | -0.14 | -0.22 | -0.03 | **-0.01** | -0.05 | -0.28 | 0.69 | 0.63 | **0.83** | 0.68 |



| | | | | | | | | | | | | |
|---|---|---|---|---|---|---|---|---|---|---|---|---|
| **Meuse** | **-0.08** | 0.15 | 0.16 | 0.26 | **-0.01** | 0.27 | 0.23 | 0.16 | 0.85 | 0.89 | **0.91** | 0.90 |
| **Kem** | -0.14 | **0.05** | -0.25 | -0.10 | 0.29 | 0.50 | **0.22** | 0.38 | 0.06 | **0.19** | 0.16 | 0.14 |
| **Kokemaenjoki** | **0.08** | 0.46 | -0.15 | -0.25 | 0.67 | 1.25 | 0.52 | **0.31** | **0.55** | 0.54 | 0.49 | 0.47 |
| **Kovda** | -0.21 | **0.04** | 0.22 | 0.10 | **1.54** | 2.44 | 2.90 | 2.54 | **0.30** | 0.27 | 0.28 | 0.29 |
| **Indalsaelven** | -0.22 | -0.19 | **-0.19** | -0.28 | **0.26** | 0.34 | 0.47 | 0.32 | 0.50 | 0.48 | **0.58** | 0.31 |
| **Luleaelven** | -0.13 | **-0.12** | -0.13 | -0.28 | 2.40 | 2.42 | 2.40 | **1.62** | 0.33 | **0.35** | 0.26 | 0.33 |





**Table 2.** River discharge scores (AC, KGE) computed at downstream GRDC gauges. For each gauge and score pair, the best/worst simulation is highlighted in bold black/grey.

| | AC | | | | KGE | | | |
|---|---|---|---|---|---|---|---|---|
| | ERA-I | P-ERA | WFDEI | PGF | ERA-I | P-ERA | WFDEI | PGF |
| Volga | **0.34** | 0.32 | 0.30 | 0.31 | -0.47 | -1.14 | -0.80 | **-0.45** |
| Danube | 0.83 | 0.85 | 0.84 | **0.86** | 0.43 | 0.06 | 0.56 | **0.70** |
| Dnepr | **0.57** | 0.57 | 0.53 | 0.57 | **-1.61** | -3.67 | -1.75 | -1.84 |
| Don | 0.29 | 0.32 | 0.30 | **0.41** | -4.92 | -10.30 | -5.62 | **-4.12** |
| Severnaya Dvina | **0.46** | 0.44 | 0.43 | 0.35 | -0.09 | 0.02 | **0.04** | -0.13 |
| Neva | 0.23 | **0.24** | 0.16 | 0.17 | **-1.05** | -1.86 | -1.07 | -1.09 |
| Vistula | 0.70 | 0.68 | **0.72** | 0.71 | 0.17 | -0.07 | 0.42 | **0.51** |
| Ural | 0.21 | 0.25 | **0.26** | 0.23 | -3.74 | -5.65 | -3.02 | **-0.99** |
| Rhine | 0.85 | **0.90** | 0.78 | 0.79 | **0.55** | 0.13 | 0.34 | 0.48 |
| Elbe | 0.71 | 0.77 | **0.80** | 0.72 | 0.46 | -0.62 | 0.59 | **0.65** |
| Oder | 0.62 | 0.64 | 0.66 | **0.66** | 0.16 | -0.50 | 0.29 | **0.35** |
| Rhone | 0.85 | **0.89** | 0.80 | 0.82 | 0.53 | 0.32 | **0.65** | 0.61 |
| Ebro | 0.69 | **0.77** | 0.71 | 0.70 | **0.69** | 0.34 | 0.51 | 0.53 |
| Nemunas | 0.63 | 0.66 | 0.62 | **0.72** | 0.17 | -0.72 | 0.15 | **0.21** |
| Po | 0.82 | **0.87** | 0.79 | 0.80 | 0.52 | 0.70 | **0.72** | 0.71 |
| Tejo | 0.76 | 0.82 | 0.81 | **0.83** | **0.83** | 0.40 | 0.72 | 0.63 |
| Daugava | 0.63 | **0.64** | 0.59 | 0.63 | 0.54 | 0.29 | 0.55 | **0.62** |
| Duero | 0.75 | **0.77** | 0.67 | 0.70 | **0.81** | -0.03 | 0.64 | 0.70 |
| Vuoksi | 0.32 | 0.33 | **0.37** | 0.35 | -1.11 | -2.02 | **-0.89** | -0.90 |
| Guadiana | 0.66 | 0.72 | **0.75** | 0.75 | 0.30 | -0.75 | 0.30 | **0.37** |
| Mezen | 0.52 | **0.53** | 0.48 | 0.43 | 0.12 | **0.29** | 0.26 | 0.03 |
| Narva | 0.48 | 0.55 | **0.57** | 0.52 | **-0.53** | -1.57 | -0.79 | -0.54 |
| Kemijoki | 0.47 | **0.47** | 0.47 | 0.39 | **0.57** | 0.55 | 0.44 | 0.24 |
| Guadalquivir | 0.69 | 0.71 | 0.72 | **0.72** | 0.43 | -1.12 | 0.28 | 0.41 |
| Goeta | 0.52 | **0.53** | 0.50 | 0.47 | **0.39** | 0.14 | 0.16 | 0.25 |
| Glama | 0.60 | 0.62 | **0.63** | 0.54 | 0.43 | 0.49 | **0.58** | 0.41 |
| Weser | 0.82 | **0.86** | 0.84 | 0.76 | 0.65 | 0.16 | 0.69 | **0.71** |
| Kymijoki | **0.53** | 0.49 | 0.51 | 0.50 | -0.78 | -1.63 | -0.64 | **-0.48** |
| Torneaelven | 0.53 | **0.55** | 0.52 | 0.48 | 0.69 | 0.62 | **0.77** | 0.52 |





| | | | | | | | | |
|---|---|---|---|---|---|---|---|---|
| **Meuse** | 0.83 | 0.89 | **0.90** | 0.87 | **0.83** | 0.67 | 0.70 | 0.68 |
| **Kem** | 0.52 | 0.53 | 0.45 | **0.55** | 0.01 | 0.05 | **0.10** | 0.05 |
| **Kokemaenjoki** | **0.62** | 0.61 | 0.56 | 0.57 | 0.19 | -0.41 | 0.26 | **0.34** |
| **Kovda** | 0.13 | 0.12 | 0.08 | **0.17** | **-0.70** | -1.55 | -2.00 | -1.64 |
| **Indalsaelven** | 0.46 | **0.46** | 0.41 | 0.40 | **0.39** | 0.35 | 0.34 | 0.19 |
| **Luleaelven** | 0.20 | 0.22 | **0.24** | 0.23 | -1.49 | -1.51 | -1.51 | **-0.78** |





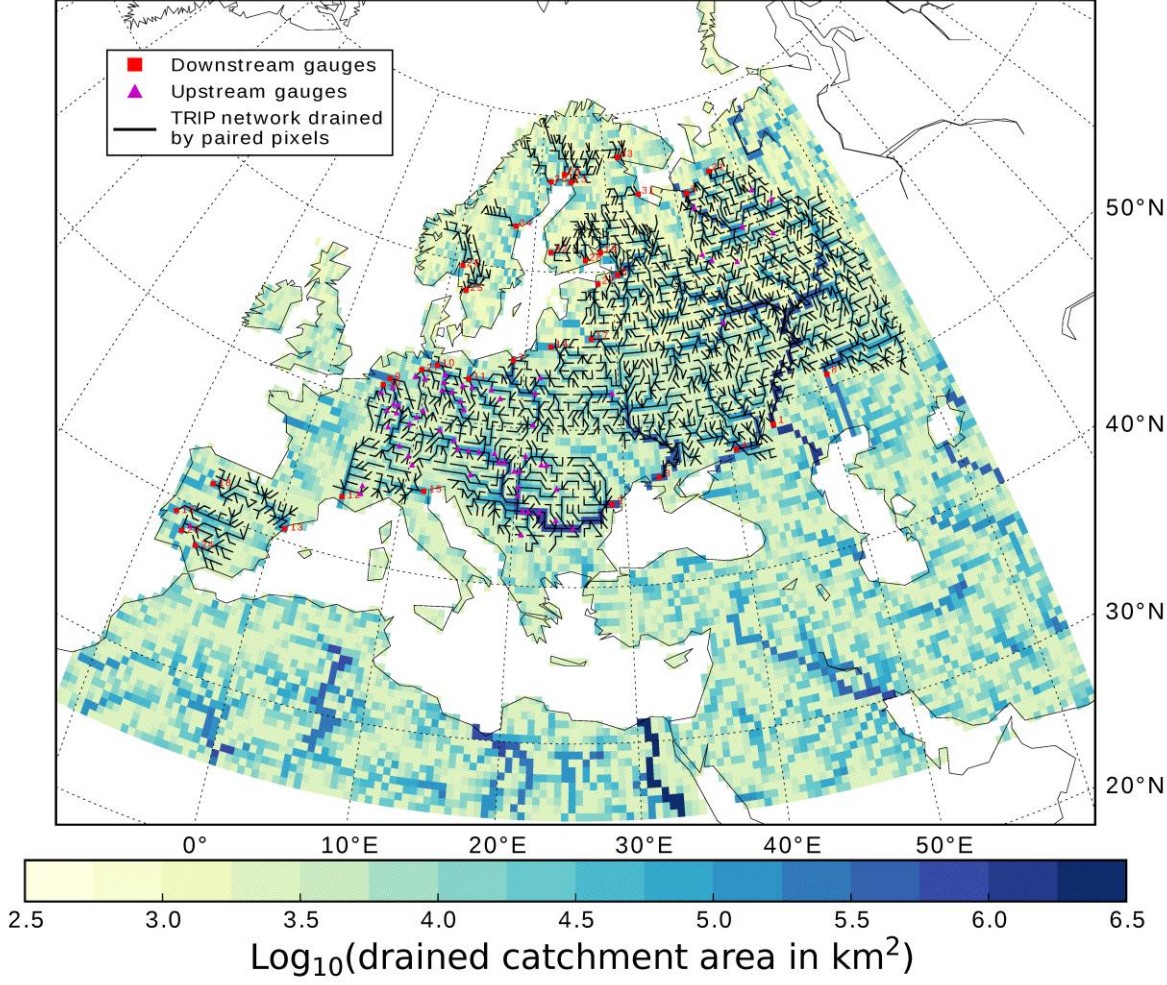

**Figure 1: Selected GRDC discharge gauges and corresponding drained CTRIP river network. The colour map categorises reflect the logarithm of the upstream catchment area of each pixel. The numbers associated to downstream gauges are sorted by descending upstream area and refer to the index N in Fig. 2. For readability, gauges are mapped on the paired pixel.**





| N | Catchment | Gauge | Area (km$^2$) | 1980 | 1985 | 1990 | 1995 | 2000 | 2005 | 2010 |
|---|-----------|-------|------|------|------|------|------|------|------|------|
| 1 | Volga | Volgograd | 1360000 | | | | | | | |
| 2 | Danube | Ceatal Izmail | 807000 | | | | | | | |
| 3 | Dnepr | Kakhovskoye V. | 482000 | | | | | | | |
| 4 | Don | Razdorskaya | 378000 | | | | | | | |
| 5 | Severnaya Dvina | Ust-Pinega | 348000 | | | | | | | |
| 6 | Neva | Novosaratovka | 281000 | | | | | | | |
| 7 | Vistula | Tczew | 194376 | | | | | | | |
| 8 | Ural | Kushum | 190000 | | | | | | | |
| 9 | Rhine | Lobith | 160800 | | | | | | | |
| 10 | Elbe | Neu-Darchau | 131950 | | | | | | | |
| 11 | Oder | Hohensaaten | 109564 | | | | | | | |
| 12 | Rhone | Beaucaire | 95590 | | | | | | | |
| 13 | Ebro | Tortosa | 84230 | | | | | | | |
| 14 | Nemunas | Smalininkai | 81200 | | | | | | | |
| 15 | Po | Pontelagoscuro | 70091 | | | | | | | |
| 16 | Tejo | Almourol | 67490 | | | | | | | |
| 17 | Daugava | Daugavpils | 64500 | | | | | | | |
| 18 | Duero | Puente Pino | 63160 | | | | | | | |
| 19 | Vuoksi | Vuoski | 61061 | | | | | | | |
| 20 | Guadiana | Pulo Do Lobo | 60883 | | | | | | | |
| 21 | Mezen | Malonisogorskaya | 56400 | | | | | | | |
| 22 | Narva | Narva | 56000 | | | | | | | |
| 23 | Kemijoki | Isohaara | 50683 | | | | | | | |
| 24 | Guadalquivir | Alcala Del Rio | 46995 | | | | | | | |
| 25 | Goeta | Vargoens | 46886 | | | | | | | |
| 26 | Glama | Langnes | 40540 | | | | | | | |
| 27 | Weser | Intschede | 37720 | | | | | | | |
| 28 | Kymijoki | Anjala | 36275 | | | | | | | |
| 29 | Torneaelven | Kukkolankoski | 33930 | | | | | | | |
| 30 | Meuse | Lith | 29000 | | | | | | | |
| 31 | Kem | Putkinskaya | 28700 | | | | | | | |
| 32 | Kokemaenjoki | Harjavalta | 26117 | | | | | | | |
| 33 | Kovda | Knyazhegubskoye | 25900 | | | | | | | |
| 34 | Indalsaelven | Bergeforsens | 25761 | | | | | | | |
| 35 | Luleaelven | Bodens | 24924 | | | | | | | |

**Figure 2: Discharge record availability and drained areas of downstream GRDC gauges. The indices N are mapped in Fig. 1.**



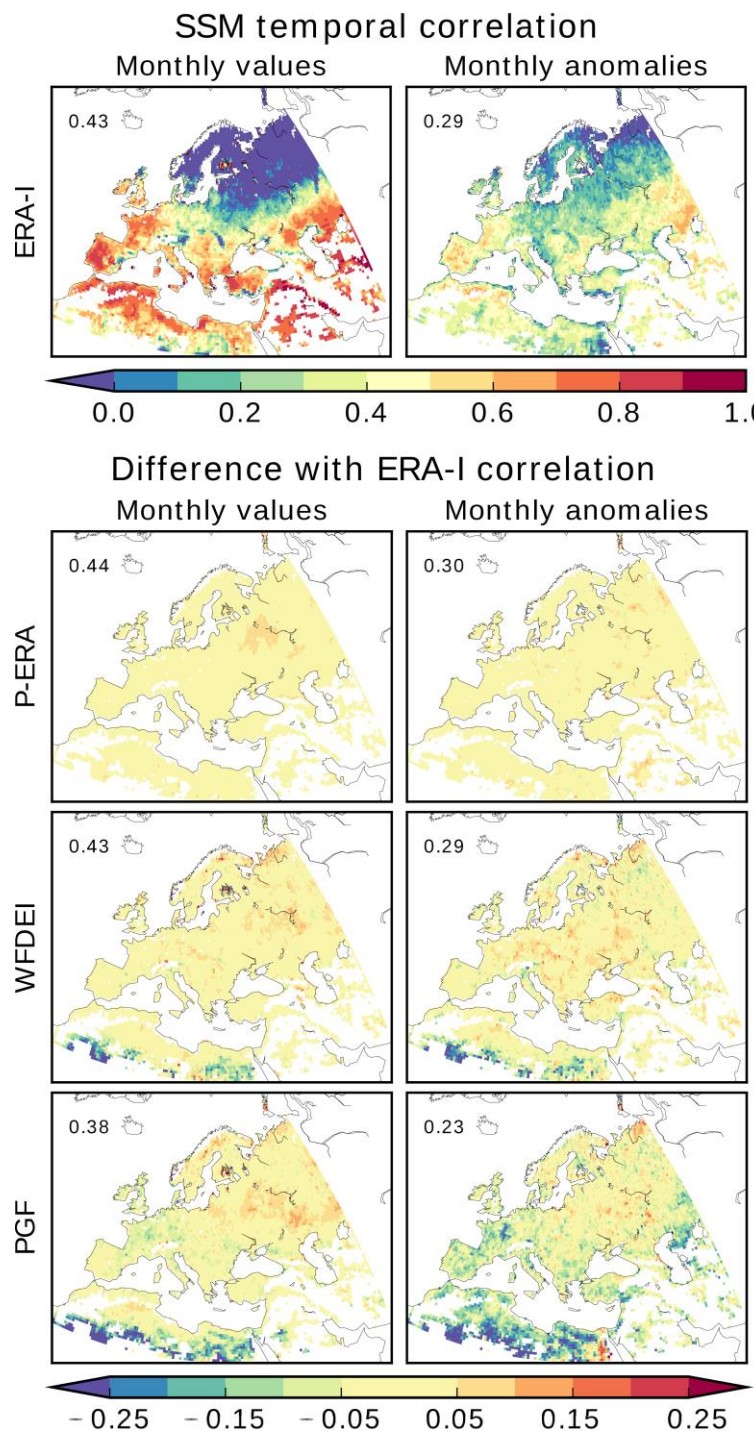

**Figure 3: Temporal correlation between modelled and remotely sensed ESA-CCI SSM for monthly averages and anomalies. Global correlation coefficients are reported in the upper left corner of each map.**



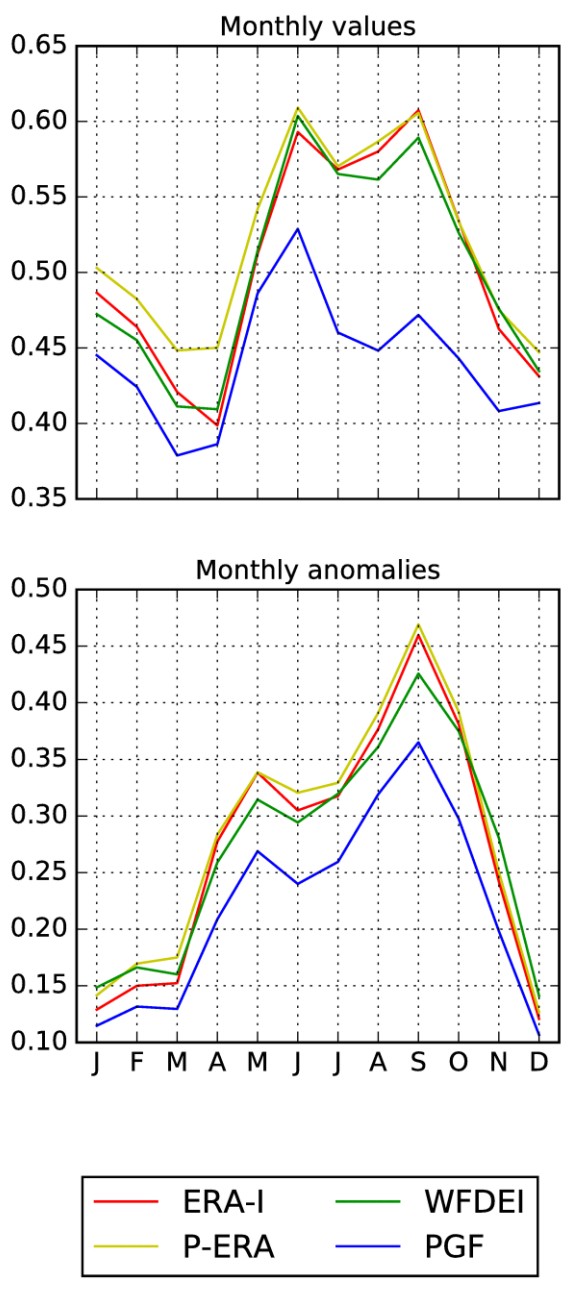

**Figure 4: Annual cycles of correlation between modelled and remotely sensed ESA-CCI SSM for monthly averages and anomalies. Coefficients are computed separately for each calendar month.**



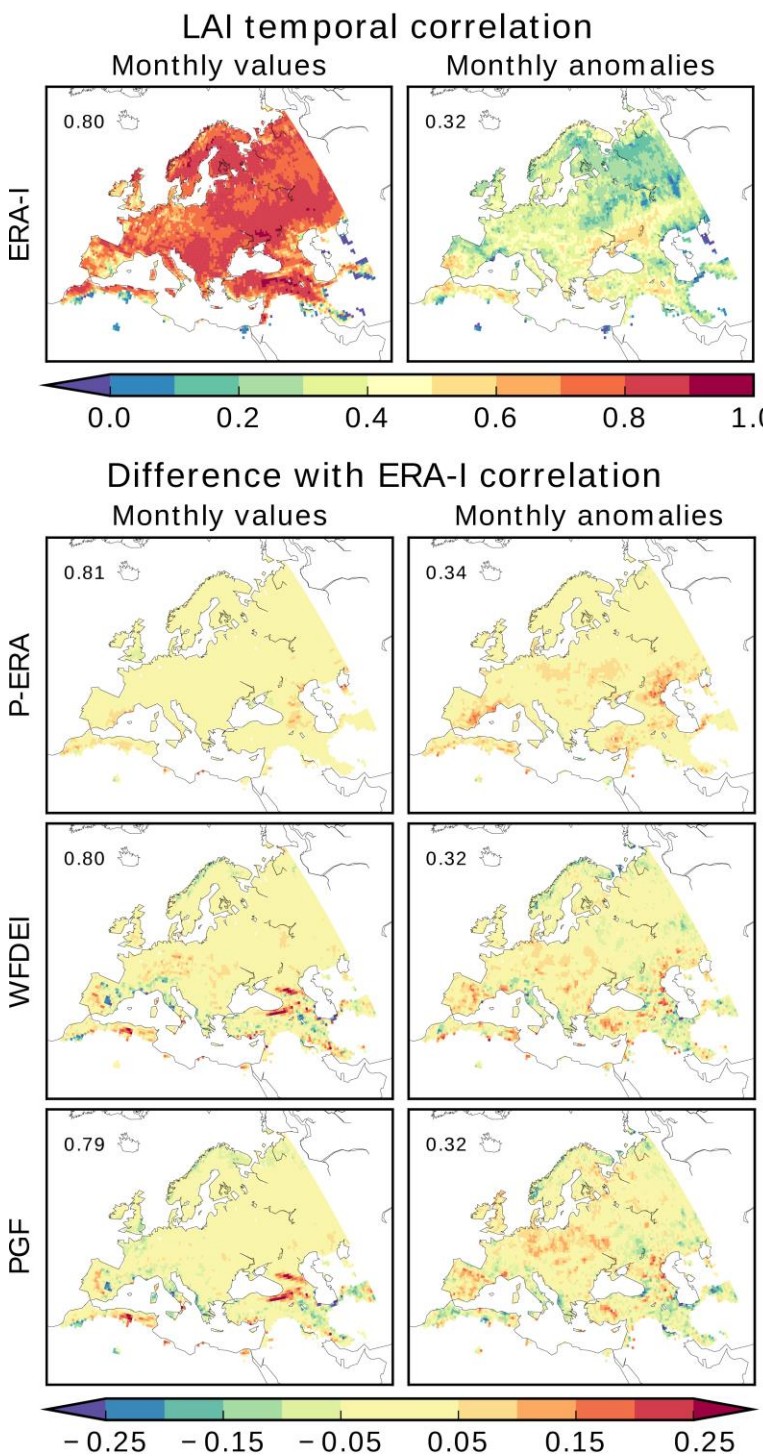

**Figure 5: Temporal correlation between modelled and remotely sensed GIMMS LAI for monthly averages and anomalies. Global correlation coefficients are reported in the upper left corner of each map.**




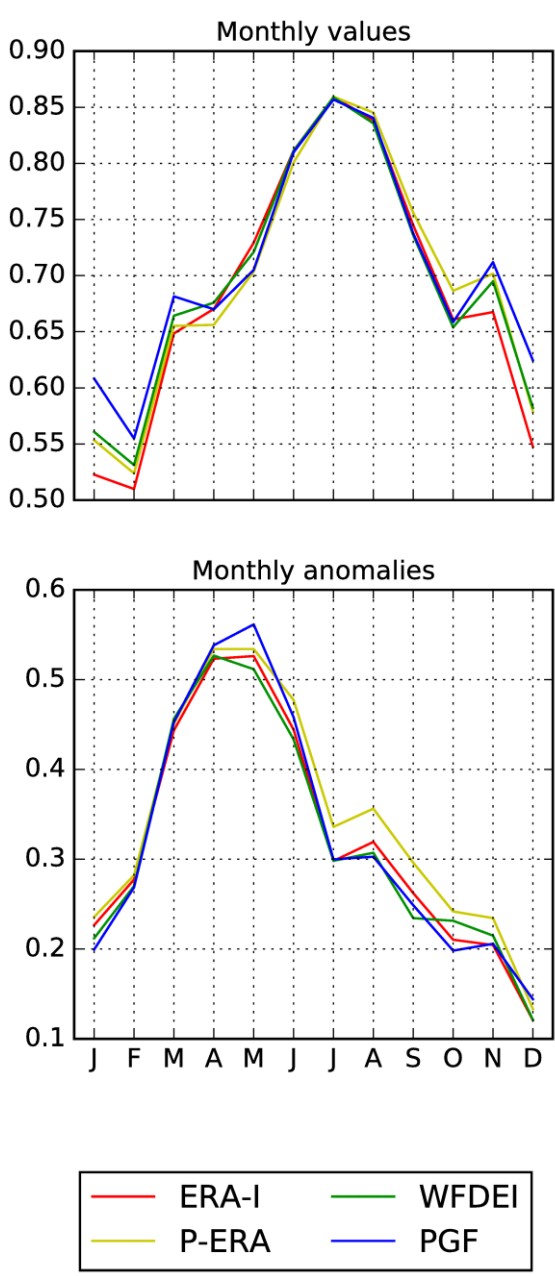

**Figure 6: Annual cycles of correlation between modelled and remotely sensed GIMMS LAI for monthly averages and anomalies. Coefficients are computed separately for each calendar month.**





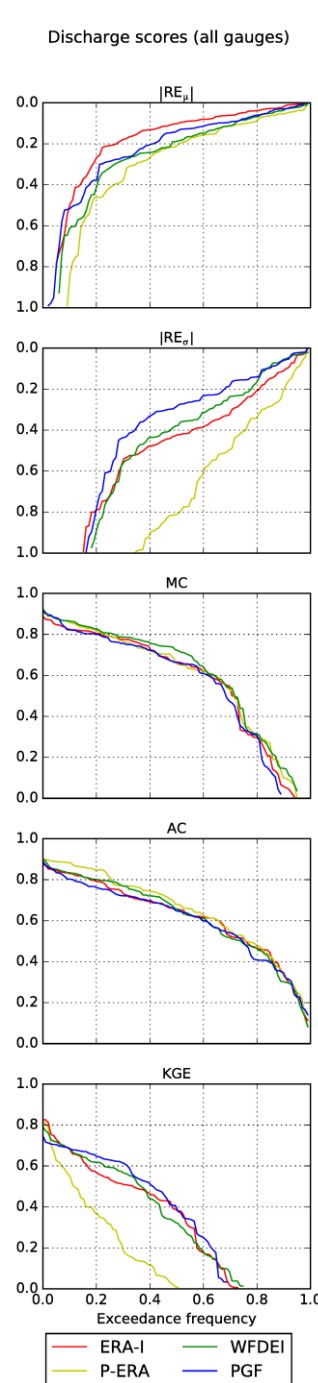

**Figure 7: Cumulated frequency plots of river discharge scores computed at all GRDC gauges. |REμ| and |REσ| values larger than 1 are not shown. MC, AC and KGE are truncated at 0.**