# Peer review of "Hydrological assessment of atmospheric forcing uncertainty in the Euro-Mediterranean area using a land surface model"

_Hydrology and Earth System Sciences, 2017_

## Referee Comment (RC1) · Anonymous Referee #1 · 11 Nov 2017

The study aims to assess the impacts of the uncertainties of atmospheric forcing to the hydrological output of a land surface model. Four different meteorological datasets were used to drive the LSM model. Outputs were validated against point river discharge measurements and remotely sensed soil moisture and leaf area index. Forcing uncertainty was quantified for all combinations of forcing and output variables.

I commend the authors for their high quality research. Their study address highly relevant research within the scope of the journal. The manuscript is well structured and the presentation of the experimental design, models and data is excellent. Considering these, my suggestion to would be to be published after addressing some minor comments:

Page 2 – Line 26:  I assume spatial scales are expressed in exponents

Page 5 – Line 22: please define the finer spatial scale.

Page 6 – Line 20: The adjustment of just one of the forcing variables (precipitation) leads to physical inconsistencies (Haddeland et al, 2012; Sippel et al, 2016). Authors could elaborate on this.

Page 7 – Line 10: Information regarding the impact of bias adjustment on large scale hydrological outputs are documented by Hagemann et al, 2011; Muerth et al, 2013; Papadimitriou et al, 2017.

Page 7 – Line 5: Replace "extrapolated" with interpolated or simply re-mapped.

Page 7 - Section: Atmospheric reference datasets. The atmospheric reference datasets that are used for comparison with the forcing are not, in some cases, independent. For example air temperature of WFDEI and PGF are bias corrected using datasets and compared against CRUv3.21. Several additional state of the art meteorogical datasets exist and could be used, like for example:

- The Berkeley Earth Surface Temperatures (BEST) (Rohde et al, 2013)
- NASA Goddard's Global Surface Temperature Analysis (GISTEMP) (Hansen et al, 2010)
- Global Historical Climatology Network (Lawrimore et al, 2011)
- Global Soil Wetness Project dataset (GSWP3) (Yoshimura and Kanamitsu, 2013)

Authors could reflect on that.

References:

Haddeland, I., Heinke, J., Voß, F., Eisner, S., Chen, C., Hagemann, S., and Ludwig, F., 2012. Effects of climate model radiation, humidity and wind estimates on hydrological simulations, Hydrol. Earth Syst. Sci., 16, 305–318, https://doi.org/10.5194/hess-16-305-2012

Hagemann, S., Chen, C., Haerter, J.O., Heinke, J., Gerten, D. and Piani, C., 2011. Impact of a statistical bias correction on the projected hydrological changes obtained from three GCMs and two hydrology models. Journal of Hydrometeorology, 12(4), pp.556-578.

Hansen, J., R. Ruedy, M. Sato, and K. Lo, 2010: Global surface temperature change, Rev. Geophys., 48, RG4004, doi:10.1029/2010RG000345

Lawrimore, J. H, M. J. Menne, B. E. Gleason, C. N. Williams, D. B. Wuertz, R. S. Vose, and J. Rennie (2011). An overview of the Global Historical Climatology Network monthly mean temperature data set, version 3, J. Geophys. Res., 116, D19121, doi:10.1029/

Muerth, M. J., Gauvin St-Denis, B., Ricard, S., Velázquez, J. A., Schmid, J., Minville, M., Caya, D., Chaumont, D., Ludwig, R., and Turcotte, R., 2013. On the need for bias correction in regional climate scenarios to assess climate change impacts on river runoff, Hydrol. Earth Syst. Sci., 17, 1189-1204, https://doi.org/10.5194/hess-17-1189-2013.

Papadimitriou, L. V., Koutroulis, A. G., Grillakis, M. G., and Tsanis, I. K. (2017). The effect of GCM biases on global runoff simulations of a land surface model, Hydrol. Earth Syst. Sci., 21, 4379-4401, https://doi.org/10.5194/hess-21-4379-2017, 2017.

Rohde R, Muller RA, Jacobsen R, Muller E, Perlmutter S, et al. (2013) A New Estimate of the Average Earth Surface Land Temperature Spanning 1753 to 2011. Geoinfor Geostat: An Overview 1:1

Sippel, S., Otto, F. E. L., Forkel, M., Allen, M. R., Guillod, B. P., Heimann, M., Reichstein, M., Seneviratne, S. I., Thonicke, K., and Mahecha, M. D., 2016. A novel bias correction methodology for climate impact simulations, Earth Syst. Dynam., 7, 71-88, https://doi.org/10.5194/esd-7-71-2016.

Yoshimura K. and M. Kanamitsu, (2013). Incremental Correction for the Dynamical Downscaling of Ensemble Mean Atmospheric Fields. Mon. Wea. Rev., 141, 3087–3101

---

## Referee Comment (RC2) · Anonymous Referee #2 · 13 Nov 2017

In the manuscript entitled "Hydrological assessment of atmospheric forcing uncertainty in the Euro-Mediterranean area using a land surface model", the authors evaluate the sensitivity of simulated top layer soil moisture, leaf area index (LAI), and streamflow by the SURFEX-CTRIP model system over Europe given five meteorological forcing data sets. The SURFEX-CTRIP model system is based on the land-surface model ISBA-A-gs, which is based on a biochemical model to simulate the interaction between the soil, biosphere, and atmosphere. The goal of the study is to assess the uncertainty in model simulation that can be attributed to the forcings. The forcing data sets are ERA-Interim reanalysis (ERA-I), ERA-I with precipitation bias-corrected to monthly values by GPCP (P-ERA), WFDEI, PGF, and a reference data set based on several observational data

sets. The authors report, in general, little impact by the different forcing data sets on bias and correlation of simulated values. Exceptions are PGF for simulated soil moisture and P-ERA for streamflow. The motivation of the study is to assess the modelling tool for planning human activities involving freshwater resources (i.e., integrated water resources management). I think this motivation is odd because normally, hydrologic models are used for this purpose instead of land surface models (LSMs). The former are designed for this purpose, they are more conceptual than the latter, and typically, parameters can be calibrated to achieve a satisfying representation of the terrestrial hydrologic cycle. On the contrary, LSMs are more physically-based, incorporate a wider range of processes (e.g., CO2-cycle), and have been developed to provide the lower boundary condition for coupled atmosphere-land-ocean models over land. Nevertheless, I think that the evaluation of a LSM modelling system by different forcing data sets is a welcome contribution to the field of hydrology, but there are several criteria that have to be met to provide a meaningful analysis. The most important criteria is that the model satisfactorilly reproduces the terrestrial hydrological cycle. Often, streamflow is used for this assessment and the authors also compare simulated streamflow against observations at 35 gauges (locations are shown in Figure 1). The median KGE over all gauges is at most 0.4 with at least 20\% of the gauges having a negative KGE, indicating a poor representation of the surface hydrology. This holds for all forcing data sets, which indicates that the poor performance is independent of these. As can be seen in the bottom row of Figure∼S1 in the supplements, most of the better performing gauges are nested sub-catchments of the Danube, Rhine, and Elbe river. These represent the same humid conditions and the good performance over the same area is "double counted". Other areas such as the cold region in North-Eastern Europe and catchments in the Mediterranean show significantly poorer performance. These also represent a large fraction of the European area. The North-Eastern part of Europe also experiences very poor agreement with soil moisture observations (Figure 3) and LAI (Figure 5). This poor agreement in this region does not allow any assessment of the forcing uncertainty because the model might lack important processes to reproduce

the terrestrial hyrdological cycle there. This might also be due to the fact that the authors use the parameters of previous work that has been only validated in the Rhone catchment (Decharme and Douville, 2006). In conclusion, it is not demonstrated that the little differences seen among the forcing data sets are not due to the chosen model parameters (e.g., soil water parameters like porosity, saturated hydraulic conductivity and d_ice etc.), or lacking hydrologic processes in the model. In other words, it is very likely that the results are an artefact of poor model performance over large parts of Europe. The authors have to investigate the sensitivity of model parameters at the different hydro-climatic regimes in Europe and choose parameters that lead to a better representation of the terrestrial hydrologic cycle before they can assess the influence of different forcing data sets. This calibration exercise should be conducted using the observation-based reference data set.

Furthermore, main conclusions are not supported by the results. I would like to give two examples for these.

1.) p. 20, l. 21 ff: The authors conclude that LSMs may progressively integrate groundwater modelling to improve the simulation of river discharge. This does not relate to the findings of this study. All the runs presented in this study include a linear reservoir to represent groundwater storage (see p. 5, l. 32f). The authors would have to present results that do not use this groundwater component to make this conclusion.

2.) p. 20, l. 24f: The authors conclude that: "Due its relevance for crop yield and water demand prediction, large-scale irrigation schemes should be embedded into LSMs". This is contradicting the findings of this study. Section 6.3 shows that the model best reproduces correlations to LAI during the maximum phenological development phase (p. 17, l. 29f). The authors further state in this section that: "This is an encouraging result for the simulation of crop yield and, in general, of the primary production of land surface ecosystems." (p. 17, l. 30f). The authors also demonstrate that LAI is systematically overestimated during the maximum phenological development. While I do agree that it is important to include irrigation in LSMs, it is not supported by these

findings. In the case of the ISBA-A-gs model, including irrigation would lower the water stress of plants during summer (i.e., the maximum phenological development phase) and lead to even more increased GPP and LAI. The authors have to conduct additional analysis and perform runs including an irrigation scheme to make this conclusion.

Overall, the authors have to either investigate which processes are lacking in ISBA-A-gs to better represent the hydrologic cycle in cold regions and semi-arid ones. Alternatively, they have to conduct a comprehensive parameter calibration study. Both of these avenues are beyond the scope of this study. Therefore, unfortunately, I have to recommend to reject this manuscript.

References:

Decharme, B., and Douville, H.: Introduction of a sub-grid hydrology in the ISBA land surface model, Climate Dynamics, 26, 1, 65–78, doi:10.1007/s00382-005-0059-7, 2005

---

## Author Comment (AC1) · 14 Dec 2017

**Gelati et al., Hydrol. Earth Syst. Sci. Discuss., https://doi.org/10.5194/hess-2017-341**

**Response to Referee #1**

Our responses to the comments of Referee #1 are organised as follows: comment 1.x from Referee #1; authors' response to 1.x and proposed changes in the manuscript, where added text is formatted italic green. Page and line numbers refer to the first submission. Added references are reported at the end of this document.

**1.1 Comment:** The study aims to assess the impacts of the uncertainties of atmospheric forcing to the hydrological output of a land surface model. Four different meteorological datasets were used to drive the LSM model. Outputs were validated against point river discharge measurements and remotely sensed soil moisture and leaf area index. Forcing uncertainty was quantified for all combinations of forcing and output variables.

I commend the authors for their high quality research. Their study address highly relevant research within the scope of the journal. The manuscript is well structured and the presentation of the experimental design, models and data is excellent. Considering these, my suggestion to would be to be published after addressing some minor comments:

**1.1 Response:** We thank Referee #1 for her/his review of the manuscript and useful comments, which we address in the following.

**1.2 Comment:** Page 2 – Line 26: I assume spatial scales are expressed in exponents

**1.2 Response:** Yes, we made a typographical error. We propose replacing "(101–102 km)" with "*$(10^1–10^2$ km)*".

**1.3 Comment:** Page 5 – Line 22: please define the finer spatial scale.

**1.3 Response:** To better characterise the spatial scales addressed by the sub-grid snow-vegetation-soil composite columns, we propose adding on page 5, line 23:

"[…] at a finer spatial scale than the one imposed by the atmospheric forcing. *In this study we use 12 columns within 0.5° latitude/longitude grid cells. The size of model grid cells is determined by the atmospheric forcing spatial resolution.*".

**1.4 Comment:** Page 6 – Line 20: The adjustment of just one of the forcing variables (precipitation) leads to physical inconsistencies (Haddeland et al, 2012; Sippel et al, 2016). Authors could elaborate on this.

**1.4 Response:** We agree this is an important aspect of bias correction. We propose adding the following paragraph in Section 6.1, on page 16, after line 20:

"*Limitations of (and alternatives to) bias correction methods adjusting a sub-set of the forcing variables or not involving statistical moments beyond the mean are discussed by Haddeland et al. (2012) and Sippel et al. (2016).*".

**1.5 Comment:** Page 7 – Line 10: Information regarding the impact of bias adjustment on large scale hydrological outputs are documented by Hagemann et al, 2011; Muerth et al, 2013; Papadimitriou et al, 2017.

**1.5 Response:** We agree on referring to recent research on the hydrological impacts of model-based atmospheric forcing bias correction. Thus, we propose adding in Section 1 ("Introduction"), on page 2, line 24:

"The experiments of Guo et al. (2006b) found that atmospheric forcing uncertainties affect LSM hydrological simulations as much as uncertainties stemming from the models themselves. *AGCM-based atmospheric forcing may be bias-corrected to better reproduce observed climatology, thus introducing an additional layer of input uncertainty (Hagemann et al., 2011; Muerth et al., 2013; Papadimitriou et al., 2017).*".

**1.6 Comment:** Page 7 – Line 5: Replace "extrapolated" with interpolated or simply re-mapped.

**1.6 Response:** We agree and propose replacing "extrapolated" with "*interpolated*".

**1.7 Comment:** Page 7 - Section: Atmospheric reference datasets. The atmospheric reference datasets that are used for comparison with the forcing are not, in some cases, independent. For example air temperature of WFDEI and PGF are bias corrected using datasets and compared against CRUv3.21. Several additional state of the art meteorogical datasets exist and could be used, like for example:

- The Berkeley Earth Surface Temperatures (BEST) (Rohde et al, 2013)

- NASA Goddard's Global Surface Temperature Analysis (GISTEMP) (Hansen et al, 2010)

- Global Historical Climatology Network (Lawrimore et al, 2011)

- Global Soil Wetness Project dataset (GSWP3) (Yoshimura and Kanamitsu, 2013)

Authors could reflect on that.

**1.7 Response:** We agree that inter-dependencies between forcing and reference atmospheric datasets may be a limit of our study. We also would like to stress that the extensive use of CRU data is motivated by its relatively high update frequency (currently updated to January 2017) and resolution (0.5°), which make this dataset attractive for land surface modelling. Moreover, since PGF and WFDEI are based on different reanalyses, their bias corrections yield different sub-diurnal variabilities. To account for these observations and to point at other existing state-of-the-art atmospheric datasets, we propose adding the following paragraph at the end of Section 6.1, on page 16, after the additional paragraph proposed in response 1.4:

"*Some of the used reference atmospheric datasets are not independent from the forcing. For example, CRU air temperature is used as reference and to bias-correct WFDEI and PGF forcing. The use of CRU data is motivated by its relatively high update frequency and resolution (0.5°), which make this dataset attractive for land surface modelling. Moreover, since PGF and WFDEI are based on different reanalyses, their bias corrections yield different sub-diurnal variabilities. However, to reduce dataset interdependencies, future work on the evaluation of forcing uncertainties may benefit from using a wider ensemble of state of the art meteorological datasets, among others: the Global Soil Wetness Project (GSWP3) forcing dataset (Yoshimura and Kanamitsu, 2013); the Goddard Institute for Space Studies Temperature (GISSTEMP) analysis (Hansen et al., 2010); the Global Historical Climatology Network temperature dataset (Lawrimore et al, 2011); The Berkeley Earth Surface Temperatures (BEST) dataset (Rohde et al., 2013); and the Multi-Source Weighted-Ensemble Precipitation (MSWEP) dataset (Beck et al., 2017b).*".

**References**

We propose citing all references suggested by Referee #1 and adding the following:

*Beck, H. E., van Dijk, A. I. J. M., Levizzani, V., Schellekens, J., Miralles, D. G., Martens, B., and de Roo, A.: MSWEP: 3-hourly 0.25° global gridded precipitation (1979–2015) by merging gauge, satellite, and reanalysis data, Hydrology and Earth System Sciences, 21, 589–615, doi:10.5194/hess-21-589-2017, 2017b.*

---

## Author Comment (AC2) · 14 Dec 2017

**Gelati et al., Hydrol. Earth Syst. Sci. Discuss., https://doi.org/10.5194/hess-2017-341**

**Response to Referee #2**

Our responses to the comments of Referee #2 are organised as follows: comment 2.x from Referee #2; authors' response to 2.x and proposed changes in the manuscript, where added text is formatted italic green. Page and line numbers refer to the first submission. Added references are reported at the end of this document.

**2.1 Comment:** In the manuscript entitled "Hydrological assessment of atmospheric forcing uncertainty in the Euro-Mediterranean area using a land surface model", the authors evaluate the sensitivity of simulated top layer soil moisture, leaf area index (LAI), and streamflow by the SURFEX-CTRIP model system over Europe given five meteorological forcing data sets. The SURFEX-CTRIP model system is based on the land-surface model ISBA-A-gs, which is based on a biochemical model to simulate the interaction between the soil, biosphere, and atmosphere. The goal of the study is to assess the uncertainty in model simulation that can be attributed to the forcings. The forcing data sets are ERA-Interim reanalysis (ERA-I), ERA-I with precipitation bias-corrected to monthly values by GPCP (P-ERA), WFDEI, PGF, and a reference data set based on several observational data.

**2.1 Response:** We thank Referee #2 for her/his review of the manuscript and useful comments, which we address in the following.

We would like to clarify that the model was forced using four atmospheric datasets. Furthermore, the observational reference datasets were used to assess the impacts of atmospheric forcing uncertainties through comparisons with model simulations (from page 3, line 30 to page 4, line 15).

**2.2 Comment:** The authors report, in general, little impact by the different forcing data sets on bias and correlation of simulated values. Exceptions are PGF for simulated soil moisture and P-ERA for streamflow.

**2.2 Response:** We regard the relatively small sensitivities to forcing uncertainty as indicators of: model robustness, if model performance is satisfactory; or model limitations (in case of unsatisfactory performance), for which we suggest several research directions in Section 6 ("Discussion"). Concerning river discharge, we find that forcing uncertainty has a larger impact on the mean and standard deviation than on the timing, shape and inter-annual variability (page 18, lines 15-25).

**2.3 Comment:** The motivation of the study is to assess the modelling tool for planning human activities involving freshwater resources (i.e., integrated water resources management). I think this motivation is odd because normally, hydrologic models are used for this purpose instead of land surface models (LSMs). The former are designed for this purpose, they are more conceptual than the latter, and typically, parameters can be calibrated to achieve a satisfying representation of the terrestrial hydrologic cycle. On the contrary, LSMs are more physically-based, incorporate a wider range of processes (e.g., CO2-cycle), and have been developed to provide the lower boundary condition for coupled atmosphere-land-ocean models over land.

**2.3 Response:** We agree that the motivations for using a land surface model (LSM) were not properly discussed.

To better frame our work in the climate change context, which requires physically consistent descriptions of water, energy and carbon cycles, we propose replacing the first sentence of the abstract (page 1, lines 10-11):

[revised manuscript text omitted]

**2.4 Comment:** Nevertheless, I think that the evaluation of a LSM modelling system by different forcing data sets is a welcome contribution to the field of hydrology, but there are several criteria that have to be met to provide a meaningful analysis. The most important criteria is that the model satisfactorilly reproduces the terrestrial hydrological cycle. Often, streamflow is used for this assessment and the authors also compare simulated streamflow against observations at 35 gauges (locations are shown in Figure 1). The median KGE over all gauges is at most 0.4 with at least 20\% of the gauges having a negative KGE, indicating a poor representation of the surface hydrology. This holds for all forcing data sets, which indicates that the poor performance is independent of these.

**2.4 Response:** Simulated streamflow is evaluated using 5 scores: the relative errors in the mean ($RE_\mu$) and standard deviation ($RE_\sigma$); the monthly (MC) and the anomaly (AC) correlation coefficients; and KGE, which aggregates the information of $RE_\mu$, $RE_\sigma$ and MC (section 4.3). While an aggregate score may be useful to simplify model calibration (by reducing it to a single-objective problem), it does not provide useful information for understanding the causes of poor model performance, which should be evaluated using multiple criteria (Gupta et al., 2009). KGE is defined as 1 minus the Euclidean distance of the components [$RE_\mu$, $RE_\sigma$, 1 - MC] from the ideal point [0, 0, 0] (Equation 2).

Therefore, KGE cannot be larger than the smallest of its components, so the "worst" score is a limiting factor. Therefore, we believe that river discharge simulations should not be evaluated only by means of KGE. The other scores should be considered as they inform about the causes of better or worse model performance.

We observe that the obtained KGE scores are strongly impacted by the relative errors, in particular by $RE_\sigma$ (page 14, lines 25-32; page 15, lines 20-22; Supplement page 1, lines 33-36; Tables 1 and 2; Figures 7 and S1). The large $RE_\sigma$ values indicate that the model tends to overestimate the amplitude of the discharge annual cycle (page 16, lines 8-10; page 18, lines 21-25; page 20, lines 7-8).

The correlation scores (MC and AC), which relate to the timing and shape of the streamflow time series, show a relatively good agreement between the simulations. In contrast, relative error scores ($RE_\mu$ and $RE_\sigma$) show larger inter-forcing spreads (page 14, lines 18-25). Thus, forcing uncertainties have the largest impacts on the river discharge mean and standard deviation (page 15, lines 22-24).

To better account for these observations, we propose:

- Adding the following paragraph after the definition of KGE (page 11, line 18): "*An aggregate score is often used as unique criterion in model calibration, because it allows applying efficient single-objective optimisation algorithms. However, model calibration is an inherently multi-objective problem (Gupta et al., 1998). Thus, aggregate scores are not as informative as their individual components for evaluating model performance (Gupta et al., 2009).*"

- Modifying the text on page 14, lines 25-27: "The relative errors *(in particular $RE_\sigma$)* have a strong impact on the summary curves of the aggregate score KGE, according to which the P-ERA simulation is dominated at almost all frequencies. Instead, a clear KGE ranking cannot be established among the other three simulations. *While KGE does not provide diagnostic information on the causes of better or worse model performance, the individual scores show that forcing uncertainties have larger impacts on the mean and standard deviation of the simulated discharge than on shape, timing and inter-annual variability.*"

**2.5 Comment:** As can be seen in the bottom row of Figure S1 in the supplements, most of the better performing gauges are nested sub-catchments of the Danube, Rhine, and Elbe river. These represent the same humid conditions and the good performance over the same area is "double counted". Other areas such as the cold region in North-Eastern Europe and catchments in the Mediterranean show significantly poorer performance. These also represent a large fraction of the European area. The North-Eastern part of Europe also experiences very poor agreement with soil moisture observations (Figure 3) and LAI (Figure 5). This poor agreement in this region does not allow any assessment of the forcing uncertainty because the model might lack important processes to reproduce the terrestrial hyrdological cycle there. This might also be due to the fact that the authors use the parameters of previous work that has been only validated in the Rhone catchment (Decharme and Douville, 2006).

**2.5 Response:** The redundant accounting of scores computed for river basins with multiple gauges is acknowledged on page 15, lines 1-5. For this reason, we also show and discuss the river discharge

scores and annual cycles at the most downstream gauges of each catchment (page 15, lines 8-24; Tables 1 and 2).

The KGE ranges of the downstream gauges of the Danube, Rhine and Elbe basins are: [0.06, 0.70], [0.13, 0.55] and [-0.62, 0.65], respectively (Tables 1 and 2). Similar ranges are spanned by the downstream gauges of other basins: Vistula [-0.07, 0.51], Rhone [0.32, 0.65], Ebro [0.34, 0.69], Po [0.52, 0.72], Tejo [0.40, 0.83], Daugava [0.29, 0.62], Duero [-0.03, 0.81], Kemijoki [0.24, 0.57], Glama [0.41, 0.58],  Weser [0.16, 0.71], Tornealven [0.52, 0.77], and Meuse [0.67, 0.83]. Some of the latter are located in North-Eastern Europe (Daugava, Kemijoki, Tornealven), in the Mediterranean area (Rhone, Ebro, Po), or in the Iberian Peninsula (Ebro, Tejo, Duero). Catchments and regions characterised by poorer model performance are discussed in Section 5.3 (page 15, lines 8-19) and in the Supplement (page 1, lines 22-36).

Small or negative temporal correlations between simulated surface soil moisture (SSM) and satellite retrievals in North-Eastern Europe are discussed on page 12, lines 11-19. We agree that the mismatches may be due to model misrepresentations of land surface processes that are particularly relevant in cold and mountainous areas. Therefore, we propose replacing the following text (page 12, lines 17-18):

"Possible causes for these low values may be vegetation density, topography and soil frost occurrences, all of which are known to negatively affect SSM retrievals."

with:

"*Possible causes for these low values may be: vegetation density, topography and soil frost occurrences, all of which are known to negatively affect SSM retrievals; and model misrepresentation of surface hydrological processes that are particularly relevant in cold and mountainous regions.*"

Several hypotheses for the satellite/model SSM mismatches and possible research directions are discussed in greater detail in Section 6.2. Concerning possible model misrepresentations of hydrological processes at high latitudes, we propose adding the following suggestion for future work (page 17, line 2):

"In light of the presented results, we suggest the following research directions: […] (Stoffelen et al., 2017)*; investigating the improvement or inclusion in LSMs of physical processes that are relevant to topsoil hydrology at high latitudes, e.g. snowmelt, flooding and ponding (Gouttevin et al., 2013)*."

LAI monthly temporal correlation coefficients in North-Eastern Europe are between 0.4 and 0.9, with most values above 0.7 (top-left map in Figure 5). In contrast, anomaly correlations  are significantly lower: all values are below 0.5 and the majority is below 0.3 (top-right map in Figure 5). Based on these values, the agreement between simulated and remotely sensed LAI in North-Eastern Europe appears consistent with the rest of the domain.

The model parameters used for this study are not those of Decharme and Douville (2006a), whom Referee #2 cites. For example, Decharme and Douville (2006a) used a 3-layer soil description, while we simulate 14 soil layers; moreover, the equations describing  the unsaturated soil water and energy fluxes are different. Our model configuration and parameters are described in Section 2 (from page 5, line 13 to page 6, line 8). The CTRIP parameters are derived from Decharme et al.

(2010), who performed a global evaluation of simulated river discharge. Land surface and vegetation parameters are provided by the ECOCLIMAP-II database (Faroux et al., 2013) and were used by Szczypta et al. (2014) and Albergel et al. (2017), who compared simulated SSM and LAI with satellite retrievals. Furthermore SURFEX-CTRIP was included in recent model intercomparison exercises (Schellekens et al., 2016; Beck et al., 2017a). Thus, we disagree with the statement "[…] the authors use the parameters of previous work that has been only validated in the Rhone catchment (Decharme and Douville, 2006)".

**2.6 Comment:** In conclusion, it is not demonstrated that the little differences seen among the forcing data sets are not due to the chosen model parameters (e.g., soil water parameters like porosity, saturated hydraulic conductivity and d_ice etc.), or lacking hydrologic processes in the model. In other words, it is very likely that the results are an artefact of poor model performance over large parts of Europe. The authors have to investigate the sensitivity of model parameters at the different hydro-climatic regimes in Europe and choose parameters that lead to a better representation of the terrestrial hydrologic cycle before they can assess the influence of different forcing data sets. This calibration exercise should be conducted using the observation-based reference data set.

**2.6 Response:** Unsatisfactory model performances that appear to be forcing-independent (thus systematic) may be due to model limitations, in addition to other causes such as forcing inter-dependence or satellite retrieval uncertainties. These are acknowledged in several parts of the manuscript, where generally we also identify possible research directions to tackle systematic model errors (page 13, lines 1-6; from page 13, line 27 to page 14, line 2; from page 16, line 22 to page 17, line 2; page 17, lines 12-19; from page 17, line 31 to page 18, line 6; from page 18, line 21 to page 19, line 16). Although this critical model evaluation aims at contributing to model improvement, we believe that the acknowledged specific limitations should not be generalised to the whole model. Moreover, SURFEX-CTRIP outputs were successfully compared to several observation datasets in previous studies (Decharme et al., 2010; Szczypta et al., 2012; Szczypta et al., 2014; Albergel et al., 2017) and included in recent model intercomparison exercises (Schellekens et al., 2016; Beck et al., 2017a), as stated in response 2.5.

Sensitivity analyses can be applied to LSMs for assessing parameter uncertainties and addressing the dimensionality of the parameter estimation problem (Bastidas et al., 2006). Since LSMs are designed to represent the complex and interconnected processes underlying the energy, water and carbon cycles, they should be evaluated by comparing a relatively large number of simulated fluxes and state variables with observations. Thus, multi-criteria (multi-objective) approaches are necessary to carry out meaningful parameter sensitivity analysis and calibration exercises (Bastidas, 1998; Gupta et al., 1998; Gupta et al., 1999; Bastidas et al., 2006). Considering the implied complexity, we believe that parameter sensitivity analysis and calibration are beyond the scope of this study. Decharme et al. (2010) performed sensitivity analyses on the groundwater delay and river flow velocity CTRIP routing parameters. Moreover, we maintain that parameter calibration is beyond the purposes of our study according the following considerations:

1. To model the energy, water and carbon cycles consistently, LSMs generally use parameters obtained from observations and measurements of land surface properties. The underlying

assumption is that uncalibrated LSMs may behave more robustly under climate and land use change scenarios than models based on pameterisations calibrated under past climate regimes and land uses. Furthermore, over-fitting a physically based model may hinder the detection of process misrepresentations and therefore model improvement (see response 2.3). Calibrated models can be good for wrong reasons (e.g. error compensation).

2. Calibrating model parameters based on several (four in our case) forcing datasets would entail: either obtaining an optimal parameter set for each forcing dataset; or defining a unique parameter set by weighting the objective function values computed for each forcing dataset. In both cases, forcing uncertainty would be translated into parameter uncertainty, thus deviating from the objectives of our study.

3. As LSM parameter calibration is an inherently multi-objective problem (Gupta et al., 1998), it would yield multiple optimal parameter sets. While this would inform about parameter sensitivity, it would also introduce parameter uncertainty in the analysis, which is out of the scope of our study.

These considerations motivate the choice of our approach. However, we are aware of the trade-off between calibrated and uncalibrated models, which we discuss in response 2.3.

**2.7 Comment:** Furthermore, main conclusions are not supported by the results. I would like to give two examples for these.

**2.7 Response:** We understand Referee #2 refers to the suggestions for model improvement that are summarised in Section 7 ("Conclusions"), on page 20, lines 17-28. These stem from the discussions of model limitations carried out in several parts of Section 6 ("Discussion"). Although our suggestions may be questionable, we do not agree with the statement "[…] main conclusions are not supported by the results", because most conclusions (from page 19, line 18 to page 20, line16) summarise the results presented in Sections 5 ("Results").

We address the specific examples provided by Referee #2 in the following responses (2.7.1 and 2.7.2).

**2.7.1 Comment:** 1.) p. 20, l. 21 ff: The authors conclude that LSMs may progressively integrate groundwater modelling to improve the simulation of river discharge. This does not relate to the findings of this study. All the runs presented in this study include a linear reservoir to represent groundwater storage (see p. 5, l. 32f). The authors would have to present results that do not use this groundwater component to make this conclusion.

**2.7.1 Response:** CTRIP uses a simplified linear reservoir to mimic the delay of soil drainage contributions to river discharge. This model component is not meant to represent actual groundwater processes (page 6, lines 1-2). We argue that including more physically-based groundwater schemes in LSMs may be beneficial for the simulation of freshwater availability.

Vergnes and Decharme (2012) showed that coupling a distributed groundwater scheme with the CTRIP river channels improved the simulations of discharge (assessed with GRDC data) and total water storage variations (assessed with GRACE satellite retrievals). To better account for these findings, we propose adding the following text on page 18, lines 32-33:

"Vergnes and Decharme (2012) implemented a simple groundwater scheme coupled to the river channel in each model grid cell allowing bidirectional water exchanges through the riverbed*, showing improvements in the simulations of total water storage variations and river discharge*.".

To clarify our suggestion in Section 7 ("Conclusions"), we propose replacing the following sentence (page 20, line 21-24):

"To improve the simulation of river discharge and to assess the impacts of water abstractions, LSMs may progressively integrate groundwater modelling (see e.g. Vergnes et al., 2014) and lake and reservoir regulation (see e.g. Hanasaki et al., 2006; Pokhrel et al., 2012)."

with:

"*To improve the simulation of freshwater availability, LSMs may progressively integrate more physically based groundwater schemes (see e.g. Vergnes and Decharme, 2012; Vergnes et al., 2014) as well as lake and reservoir regulation (see e.g. Hanasaki et al., 2006; Pokhrel et al., 2012).*"

**2.7.2 Comment:** 2.) p. 20, l. 24f: The authors conclude that: "Due its relevance for crop yield and water demand prediction, large-scale irrigation schemes should be embedded into LSMs". This is contradicting the findings of this study. Section 6.3 shows that the model best reproduces correlations to LAI during the maximum phenological development phase (p. 17, l. 29f). The authors further state in this section that: "This is an encouraging result for the simulation of crop yield and, in general, of the primary production of land surface ecosystems." (p. 17, l. 30f). The authors also demonstrate that LAI is systematically overestimated during the maximum phenological development. While I do agree that it is important to include irrigation in LSMs, it is not supported by these findings. In the case of the ISBA-A-gs model, including irrigation would lower the water stress of plants during summer (i.e., the maximum phenological development phase) and lead to even more increased GPP and LAI. The authors have to conduct additional analysis and perform runs including an irrigation scheme to make this conclusion.

**2.7.2 Response:** We agree that the inclusion of irrigation schemes in LSMs deserves a complex discussion that is beyond the scope of this study. Therefore we propose deleting the following sentence from Section 7 ("Conclusions"), on page 20, lines 24-25: "Due its relevance for crop yield and water demand prediction, large-scale irrigation schemes should be embedded into LSMs (Jägermeyr et al., 2016).".

**2.8 Comment:** Overall, the authors have to either investigate which processes are lacking in ISBA-A-gs to better represent the hydrologic cycle in cold regions and semi-arid ones. Alternatively, they

have to conduct a comprehensive parameter calibration study. Both of these avenues are beyond the scope of this study. Therefore, unfortunately, I have to recommend to reject this manuscript.

**2.8 Response:** We understand this last comment of Referee #2 summarises previous ones, which we address in detail in the corresponding responses. Here we report the main points of the latter.

Regarding the misrepresented or lacking physical processes in the model, our work contributes to their identification with constructive suggestions (see responses 2.2, 2.5, 2.6 and 2.7). However, we believe that further work aimed at rigorously testing the inclusion of new process representations is beyond the scope of this study, which focuses on the assessment of the hydrological impacts of atmospheric forcing uncertainties.

SURFEX-CTRIP simulations have already been successfully compared to observations/retrievals of river discharge, surface soil moisture and leaf area index either globally or in the Euro-Mediterranean area by previous studies (Decharme et al., 2010; Szczypta et al., 2012; Szczypta et al., 2014; Albergel et al., 2017). Moreover, SURFEX-CTRIP has been included in recent global model intercomparison exercises (Schellekens et al., 2016; Beck et al., 2017a), together with other state-of-the-art land surface and global hydrological models (see responses 2.5 and 2.6).

The reasons for choosing an uncalibrated physically-based modelling approach, as well as those for excluding parameter calibration from the scope of this study, are thoroughly discussed in responses 2.3 and 2.6.

---

## Author Response (AR1)

**"Hydrological assessment of atmospheric forcing uncertainty in the Euro-Mediterranean area using a land surface model"**
**by Gelati et al.**

**Cover letter to the editor**

20 February 2018

Dear Dr. Matthias Bernhardt,

The authors' response to the comments of the two anonymous referees was published on the HESS web site.

All changes relative to the discussion paper are detailed in the marked-up version of the new manuscript. They include all the response elements given by the authors in response to the reviewers' comments (green and blue for Reviewer 1 and 2, respectively).

Yours sincerely,

Jean-Christophe Calvet, Emiliano Gelati.

**LIST OF CHANGES MADE IN RESPONSE TO COMMENTS OF REVIEWER #1**

**1.3 [Page 5 – Line 22: please define the finer spatial scale.]**

**Response 1.3:** changes in the marked-up version of the revised manuscript

**P. 6, L. 20-20:**
"In this study we use 12 columns within 0.5° latitude/longitude grid cells. The size of model grid cells is determined by the atmospheric forcing spatial resolution."

**1.4 [Page 6 – Line 20: The adjustment of just one of the forcing variables (precipitation) leads to physical inconsistencies (Haddeland et al, 2012; Sippel et al, 2016). Authors could elaborate on this.]**

**Response 1.4:** changes in the marked-up version of the revised manuscript

**P. 17, L. 30-31:**
"Limitations of (and alternatives to) bias correction methods adjusting a sub-set of the forcing variables or not involving statistical moments beyond the mean are discussed by Haddeland et al. (2012) and Sippel et al. (2016)."

**1.5 [Page 7 – Line 10: Information regarding the impact of bias adjustment on large scale hydrological outputs are documented by Hagemann et al, 2011; Muerth et al, 2013; Papadimitriou et al, 2017.]**

**Response 1.5:** changes in the marked-up version of the revised manuscript

**P. 3, L. 20-22:**
"AGCM-based atmospheric forcing may be bias-corrected to better reproduce observed climatology, thus introducing an additional layer of input uncertainty (Hagemann et al., 2011; Muerth et al., 2013; Papadimitriou et al., 2017)."

**1.7 [Page 7 - Section: Atmospheric reference datasets. The atmospheric reference datasets that are used for comparison with the forcing are not, in some cases, independent. For example air temperature of WFDEI and PGF are bias corrected using datasets and compared against CRUv3.21. Several additional state of the art meteorogical datasets exist and could be used, like for example:**
**·        The Berkeley Earth Surface Temperatures (BEST) (Rohde et al, 2013)**
**·        NASA Goddard's Global Surface Temperature Analysis (GISTEMP) (Hansen et al, 2010)**
**·        Global Historical Climatology Network (Lawrimore et al, 2011)**
**·        Global Soil Wetness Project dataset (GSWP3) (Yoshimura and Kanamitsu, 2013)**
**Authors could reflect on that.]**

**Response 1.7:** changes in the marked-up version of the revised manuscript

**P. 18, L. 1-10:**

"Some of the used reference atmospheric datasets are not independent from the forcing. For example, CRU air temperature is used as reference and to bias-correct WFDEI and PGF forcing. The extensive use of CRU data is motivated by its relatively high update frequency and resolution (0.5°), which make this dataset attractive for land surface modelling. Moreover, since PGF and WFDEI are based on different reanalyses, their bias corrections yield different sub-diurnal variabilities. However, to reduce dataset interdependencies, future work on the evaluation of forcing uncertainties may benefit from using a wider ensemble of state of the art meteorological datasets, among others: the Global Soil Wetness Project (GSWP3) forcing dataset (Yoshimura and Kanamitsu, 2013); the Goddard Institute for Space Studies Temperature (GISSTEMP) analysis (Hansen et al., 2010); the Global Historical Climatology Network temperature dataset (Lawrimore et al, 2011); The Berkeley Earth Surface Temperatures (BEST) dataset (Rohde et al., 2013); and the Multi-Source Weighted-Ensemble Precipitation (MSWEP) dataset (Beck et al., 2017b)."

**LIST OF CHANGES MADE IN RESPONSE TO COMMENTS OF REVIEWER #2**

**2.3 [The motivation of the study is to assess the modelling tool for planning human activities involving freshwater resources (i.e., integrated water resources management). I think this motivation is odd because normally, hydrologic models are used for this purpose instead of land surface models (LSMs). The former are designed for this purpose, they are more conceptual than the latter, and typically, parameters can be calibrated to achieve a satisfying representation of the terrestrial hydrologic cycle. On the contrary, LSMs are more physically-based, incorporate a wider range of processes (e.g., CO2-cycle), and have been developed to provide the lower boundary condition for coupled atmosphere-land-ocean models over land.]**

**Response 2.3:** changes in the marked-up version of the revised manuscript

**P. 1, L. 10-11:**
"Physically consistent descriptions of land surface hydrology are crucial for planning human activities that involve freshwater resources, especially in light of the expected climate change scenarios."

[revised manuscript text omitted]

**2.4 [Nevertheless, I think that the evaluation of a LSM modelling system by different forcing data sets is a welcome contribution to the field of hydrology, but there are several criteria that have to be met to provide a meaningful analysis. The most important criteria is that the model satisfactorilly reproduces the terrestrial hydrological cycle. Often, streamflow is used for this assessment and the authors also compare simulated streamflow against observations at 35 gauges (locations are shown in Figure 1). The median KGE over all gauges is at most 0.4 with at least 20\% of the gauges having a negative KGE, indicating a poor representation of the surface hydrology. This holds for all forcing data sets, which indicates that the poor performance is independent of these.]**

**Response 2.4: changes in the marked-up version of the revised manuscript**

**P. 12, L. 15-18:**
"An aggregate score is often used as unique criterion in model calibration, because it allows applying efficient single objective optimisation algorithms. However, model calibration is an inherently multi-objective problem (Gupta et al., 1998). Thus, aggregate scores are not as informative as their individual components for evaluating model performance (Gupta et al., 2009)."

**P. 15, L. 31 to P. 16, L. 2:**
"While KGE does not provide diagnostic information on the causes of better or worse model performance, the individual scores show that forcing uncertainties have larger impacts on the mean and standard deviation of the simulated discharge than on shape, timing and inter-annual variability."

**2.5 [As can be seen in the bottom row of Figure S1 in the supplements, most of the better performing gauges are nested sub-catchments of the Danube, Rhine, and Elbe river. These represent the same humid conditions and the good performance over the same area is "double counted". Other areas such as the cold region in North-Eastern Europe and catchments in the Mediterranean show significantly poorer performance.**

**These also represent a large fraction of the European area. The North-Eastern part of Europe also experiences very poor agreement with soil moisture observations (Figure 3) and LAI (Figure 5). This poor agreement in this region does not allow any assessment of the forcing uncertainty because the model might lack important processes to reproduce the terrestrial hyrdological cycle there. This might also be due to the fact that the authors use the parameters of previous work that has been only validated in the Rhone catchment (Decharme and Douville, 2006).]**

**Response 2.5:** changes in the marked-up version of the revised manuscript

**P. 13, L. 21-23:**
"Possible causes for these low values may be: vegetation density, topography and soil frost occurrences, all of which are known to negatively affect SSM retrievals; and model misrepresentation of surface hydrological processes that are particularly relevant in cold and mountainous regions."

**P. 18, L. 21-22:**
"... investigating the improvement or inclusion in LSMs of physical processes that are relevant to topsoil hydrology at high latitudes, e.g. snowmelt, flooding and ponding (Gouttevin et al., 2013)."

**2.7.1 [p. 20, l. 21 ff: The authors conclude that LSMs may progressively integrate groundwater modelling to improve the simulation of river discharge. This does not relate to the findings of this study. All the runs presented in this study include a linear reservoir to represent groundwater storage (see p. 5, l. 32f). The authors would have to present results that do not use this groundwater component to make this conclusion.]**

**Response 2.7.1:** changes in the marked-up version of the revised manuscript

**P. 20, L. 20-21:**
"*Vergnes and Decharme (2012) implemented a simple groundwater scheme coupled to the river channel in each model grid cell allowing bidirectional water exchanges through the riverbed*, showing improvements in the simulations of total water storage variations and river discharge."

**P. 22, L. 8-11:**

[revised manuscript text omitted]